# Multi-modal Gaussian Process Variational Autoencoders for Neural and Behavioral Data

**Rabia Gondur**\*
Fordham University
rgondur@fordham.edu

**Usama Bin Sikandar**
Georgia Institute of Technology
usama@gatech.edu

**Evan S. Schaffer**
Icahn School of Medicine at Mount Sinai
evan.schaffer@mssm.edu

**Mikio Aoi**
University of California, San Diego
maoi@ucsd.edu

**Stephen Keeley**
Fordham University
skeeley1@fordham.edu

## Abstract

Characterizing the relationship between neural population activity and behavioral data is a central goal of neuroscience. While latent variable models (LVMs) are successful in describing high-dimensional time-series data, they are typically only designed for a single type of data, making it difficult to identify structure shared across different experimental data modalities. Here, we address this shortcoming by proposing an unsupervised LVM which extracts temporally evolving shared and independent latents for distinct, simultaneously recorded experimental modalities. We do this by combining Gaussian Process Factor Analysis (GPFA), an interpretable LVM for neural spiking data with temporally smooth latent space, with Gaussian Process Variational Autoencoders (GP-VAEs), which similarly use a GP prior to characterize correlations in a latent space, but admit rich expressivity due to a deep neural network mapping to observations. We achieve interpretability in our model by partitioning latent variability into components that are either shared between or independent to each modality. We parameterize the latents of our model in the Fourier domain, and show improved latent identification using this approach over standard GP-VAE methods. We validate our model on simulated multi-modal data consisting of Poisson spike counts and MNIST images that scale and rotate smoothly over time. We show that the multi-modal GP-VAE (MM-GPVAE) is able to not only identify the shared and independent latent structure across modalities accurately, but provides good reconstructions of both images and neural rates on held-out trials. Finally, we demonstrate our framework on two real-world multi-modal experimental settings: *Drosophila* whole-brain calcium imaging alongside tracked limb positions, and *Manduca sexta* spike train measurements from ten wing muscles as the animal tracks a visual stimulus.

## 1 Introduction

Recent progress in experimental neuroscience has enabled researchers to record from a large number of neurons while animals perform naturalistic behaviors in sensory-rich environments. An important prerequisite to analyzing these data is to identify how the high-dimensional neural data is related to the corresponding behaviors and environmental settings. Traditionally, researchers often employ a two-step approach, involving first dimensionality reduction on neural data followed by a post-hoc investigation of the latent structure of neural recordings with respect to experimental variables of interest such as stimuli or behavior (Wu et al., 2017; Zhao and Park, 2017; Saxena and Cunningham,

---

\*Currently affiliated with Cold Spring Harbor Laboratory

2019; Bahg et al., 2020; Pei et al., 2021). However, recent deep learning advancements allow for both experimental and behavioral variables to be part of a single latent-variable model (Schneider et al., 2023; Hurwitz et al., 2021; Sani et al., 2021; Singh Alvarado et al., 2021), thus opening a new avenue for unsupervised discoveries on the relationship between neural activity and behavior.

The existing approaches that jointly model neural activity and behavior are limited in that they often rely on a single latent space to describe data from both modalities (Schneider et al., 2023; Hurwitz et al., 2021), making it difficult for practitioners to isolate latent features that are shared across and independent to neural activity or behavior. Approaches that do isolate modality specific and shared latent structure either do so with no temporal structure (Gundersen et al., 2019; Kleinman et al., 2024; Singh Alvarado et al., 2021; Wu and Goodman, 2018; Shi et al., 2019; Brenner et al., 2022), or use a relatively inflexible linear dynamical system (Sani et al., 2021). Moreover, because of the deep neural network mapping from latents to observations, many existing multi-modal approaches generate latent spaces that are not obviously related to experimental variables of interest, and so additional model features are often added to aid in interpretability and analysis in neuroscience settings (Schneider et al., 2023; Hurwitz et al., 2021; Zhou and Wei, 2020).

However, LVMs developed for neural data often are able to uncover interpretable features using neural activity alone in an unsupervised fashion with minimal *a priori* assumptions. One example of these is Gaussian Process Factor Analysis (GPFA), a widely used LVM in neuroscience that finds smoothly evolving latent dynamics in neural population activity, and can illustrate different aspects of neural processing (Yu et al., 2008; Duncker and Sahani, 2018; Keeley et al., 2020a;b; Balzani et al., 2022). GPFA constrains the latent space of neural activity through the use of a Gaussian Process (GP) prior. GP priors have also been adapted to regularize the latent space of variational autoencoders (GP-VAE or GP-prior VAE) with a variety of applications (Casale et al., 2018; Fortuin et al., 2020; Ramchandran et al., 2021; Yu et al., 2022). In each of these approaches, the GP prior provides a flexible constraint in the latent dimension, specifying correlations across auxiliary observations like viewing angle, lighting, or time. The GP prior often is used for out-of-sample prediction in GP-VAEs, but in GPFA is frequently used to visualize latent structure in noisy neural population activity on a trial-by-trial basis (Duncker and Sahani, 2018; Zhao and Park, 2017; Keeley et al., 2020a). Here, we wish to leverage the interpretability seen in unsupervised GPFA models with the power of GP-VAEs to use with multi-modal time-series datasets in neuroscience.

We propose a model for jointly observed neural and behavioral data that partitions shared and independent latent subspaces across data modalities while flexibly preserving temporal correlations using a GP prior. We call this the multi-modal Gaussian Process variational autoencoder (MM-GPVAE). Our first innovation is to parameterize the latent space time-series in terms of a small number of Fourier frequencies, an approach that has been used before in the linear GPFA setting (Paciorek, 2007; Aoi and Pillow, 2017; Keeley et al., 2020a). We show that the Fourier representation dramatically improves latent identification for the standard GP-VAE. Our second innovation is to augment our new Fourier GP-VAE model to describe two data modalities simultaneously by mapping the latents linearly to neural data (like in GPFA), as well as nonlinearly to another experimental variable via a deep neural network (like a GP-VAE). We leave the specific identity of this other experimental modality intentionally vague - it could be keypoints of limb positions as animals freely move or visual stimuli across time. Because this observation modality is characterized by a deep neural network, our model can be adapted to any experimental variable of interest.

We validate our MM-GPVAE model on a simulated dataset of a smoothly rotating and scaling MNIST digit alongside simulated Poisson neural activity, whereby the digit data and the spiking data share latent structure. We show that the MM-GPVAE model is able to recover both shared and independent latent structures while simultaneously providing accurate reconstructions of data from both modalities. Lastly, we demonstrate the utility of our model by fitting the MM-GPVAE to two real-world multi-modal neural datasets: 1) *Drosophila* (fly) whole-brain calcium imaging alongside tracked 16 limb positions, and 2) *Manduca sexta* (hawkmoth) spike train measurements from ten wing muscles alongside a continuously moving visual stimulus. In the former case, we are able to view *Drosophila* behavioral conditions across shared and independent subspaces and in the latter case, we show distinct time-varying latent components tracking muscle and stimuli movement in the experiment. By showing the MM-GPVAE in these two domains, we demonstrate that our model is adaptable to a range of diverse experimental preparations in systems neuroscience.

## 2 THE GAUSSIAN PROCESS VARIATIONAL AUTOENCODER

The Gaussian Process variational autoencoder (GPVAE) uses high-dimensional data (e.g. images) accompanied by auxiliary information, like viewing angle, lighting, object identity, or observation time. This auxiliary information provides the indices in the GP prior latent representation, specifying correlations across the latent space and allowing for out-of-sample predictions at new auxiliary values (Casale et al., 2018; Fortuin et al., 2020; Ramchandran et al., 2021). However, in our case we consider continuously observed time-series data in experimental neuroscience experiments, so our auxiliary information here is evenly sampled time-bins. Because of this, we can leverage the advantages of the Fourier-domain GP representation (Aoi and Pillow, 2017; Keeley et al., 2020b; Paciorek, 2007).

Formally, consider smoothly-varying image data across timepoints, represented by the pixels-by-time matrix $\mathbf{Y} \in \mathbb{N}^{N \times T}$. For latent variable $\mathbf{z}(t) \in \mathbb{R}^P$ each latent $z_p(t)$ ($t \in \{1, 2 \ldots T\}$) evolves according to a Gaussian process, $z_p(t) \sim \mathcal{GP}(0, k_\theta(\cdot, \cdot))$, with covariance kernel $k_\theta$. The time-by-time covariance matrix of each $z_p(t)$ is then given by the Gram matrix $\mathbf{K}_\theta$ corresponding to $k_\theta$. We use a squared exponential (RBF) kernel for $\mathbf{K}$ governed by a marginal variance and length scale $\theta = \{\rho, \ell\}$ with an additive diagonal term $\alpha \mathbf{I}$ to help with inference (Yu et al., 2009; Casale et al., 2018). The likelihood of the image data is Gaussian with mean given by the latent values at any timepoint $t$ passed through a deep neural network $g_\psi(\cdot)$ with parameters $\psi$ and whose covariance is $\sigma_y^2 \mathbf{I}$.

The GP-VAE is learned using standard VAE amortized inference (Kingma and Welling, 2013), where the parameters $\mu_\phi$ and $\sigma_\phi^2$ of a variational distribution $q_\phi$ are given as neural network functions of the observed data $\mathbf{Y}$, parameterized by $\phi$. Here, the evidence lower bound (ELBO) is maximized with respect to the variational parameters $\phi$, model parameters $\psi$, and GP hyperparameters. In this work, $\alpha$ is set to a fixed value of $1e-2$ for all experiments except for the final data analysis example, where it is set to a value of $1e-4$. While the ELBO may be expressed in a variety of ways, we will follow the form that includes the variational entropy term. For details about this approach for the standard GP-VAE, see Casale et al. (2018).

### 2.1 FOURIER-DOMAIN REPRESENTATION OF THE GP-VAE

We consider a version of the GP-VAE whose auxiliary variables are a Fourier frequency representation of the time domain, as opposed to timepoints sampled on a regular lattice. This allows us to parameterize a frequency representation of the latent variables that is Gaussian distributed according to $\tilde{\mathbf{z}}_p \sim \mathcal{N}(0, \tilde{\mathbf{K}}_\theta)$ where the original RBF GP covariance matrix $\mathbf{K}$ may be diagonalized by $\tilde{\mathbf{K}} = \mathbf{B}\mathbf{K}\mathbf{B}^\top$ Here, $\mathbf{B}$ is the orthonormal discrete Fourier transform matrix and $\tilde{\mathbf{z}}_p$ is a single Fourier-domain latent vector whose length is equal to $F$, the total number of Fourier frequencies representing the latent. The model prior can now be written as

$$p(\tilde{\mathbf{Z}} \mid \boldsymbol{\theta}, \alpha) = \prod_{p=1}^{P} \mathcal{N}\left(\tilde{\mathbf{z}}_p \mid \mathbf{0}, \tilde{\mathbf{K}}_\theta + \alpha \mathbf{I}_F\right) \tag{1}$$

Where $\tilde{\mathbf{Z}}$ represents a $P \times F$ matrix of the frequency representation of the $P$ latent variables at the $F$ frequencies and $\tilde{\mathbf{K}}_\theta$ is an $F \times F$ covariance matrix. The model likelihood retains the same form as the standard GP-VAE,

$$p(\mathbf{Y}|\mathbf{Z}, \psi, \sigma^2) = \prod_{t=1}^{T} \mathcal{N}(g_\psi(\mathbf{z}_t), \sigma^2 \mathbf{I}_T) \tag{2}$$

where $\mathbf{z}_t$ represents the $t^{\text{th}}$ time index of $\mathbf{Z}$, a $P \times T$ matrix where each row is the time-domain representation of the $p^{\text{th}}$ latent, given by $\mathbf{z}_p = \mathbf{B}\tilde{\mathbf{z}}_p$.

We define the variational posterior so that it factorizes over the Fourier frequencies. The variational distribution can be written as

$$q_\phi(\tilde{\mathbf{Z}}^i \mid \mathbf{Y}^i) = \prod_{\omega} \mathcal{N}\left(\tilde{\mathbf{z}}_\omega^i \mid \tilde{\boldsymbol{\mu}}_\phi\left(\mathbf{Y}^i\right), \operatorname{diag}\left(\tilde{\boldsymbol{\sigma}}_\phi^2\left(\mathbf{Y}^i\right)\right)\right), \tag{3}$$

where $i$ indexes a trial of times-series images $\mathbf{Y}$ and $\omega$ indexes the Fourier frequencies ($\omega \in 0, 1, 2 \ldots F$). In contrast to the standard GP-VAE (Casale et al., 2018), the Fourier representation

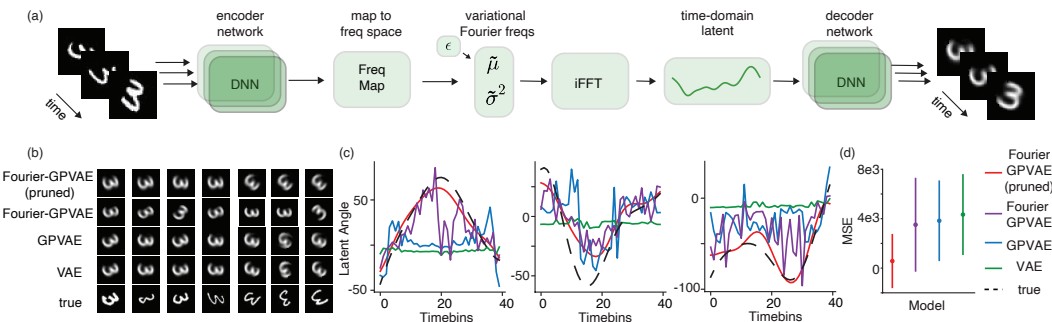

Figure 1: (a) schematic for the Fourier domain GP-VAE. All images at all timepoints for a given trial are encoded via a deep neural network into variational parameters of a pruned Fourier representation of the latent space. This Fourier representation is then mapped back into the time domain before being passed through a decoder network to give the image reconstruction at each timepoint. (b) Image reconstructions of the standard VAE, GP-VAE and Fourier domain GP-VAE. (c) Estimated latent for each model alongside the true underlying latent angle. (d) Mean squared error (MSE) of estimated latents and true latents for 60 held-out trials. Error bars indicate standard error.

requires that, for each trial, images at all timepoints are mapped to a single $P \times F$-dimensional Fourier representation. We accomplish this in two steps - first, we use a deep neural network for each image at each time point $\mathbf{y}_t^i$, which will result in $T$ total network embeddings for a single trial, each of dimension $P$. We follow that with a single linear layer each for the mean and variance of the variational distribution, $l_{\tilde{\mu}}(\cdot), l_{\tilde{\sigma}^2}(\cdot)$, that will map from the number of timepoints to the number of Fourier frequencies. $l_{\tilde{\mu}}, l_{\tilde{\sigma}^2} : I\!R^{P \times T} \to I\!R^{P \times F}$. A schematic for the Fourier domain variational and generative architecture is displayed in Figure 1(a).

There are a number of advantages to representing the GP-VAE latents and variational parameters in the Fourier domain. For one, the diagonal representation of $\hat{K}$ means we avoid a costly matrix inversion when evaluating the GP prior (Aoi and Pillow, 2017; Paciorek, 2007; Hensman et al., 2017; Wilson and Adams, 2013; Loper et al., 2021). Secondly, we can *prune* the high frequencies in the Fourier domain, effectively sparsifying the variational parameters, and hence frequencies ($F < T$), while enforcing a smooth latent representation. Lastly, the Fourier representation of the variational parameters mean we can retain both temporal correlations as well as the advantages of using a mean-field approximation for the variational posterior $q_\phi$. These advantages have been seen in simpler GP models. For more information, see (Aoi and Pillow, 2017; Keeley et al., 2020a).

The Fourier-represented GP-VAE dramatically improves the ability of the GP-VAE to learn a true underlying smoothly evolving latent for high-dimension data in a non-linear model. We demonstrate this on a simulated example using an MNIST digit that rotates by a time-varying angle given by a draw from a GP with an RBF kernel. We fit our Fourier GP-VAE model as well as the standard VAE and standard GP-VAE models to these trials. In each case, we are looking to recover the true underlying generative latent angle and hence we assert a one-dimensional latent dimensionality for the model. For more information on training and testing, see appendix.

Because the Fourier variational distribution preserves temporal correlations and prunes the high-frequency components of the latents $\mathbf{z}$, the Fourier GP-VAE model shows much smoother underlying latent representations on held-out trials (Fig 1 (c)), though each model retains the ability to accurately reconstruct the images through the network mapping from the latent space (Fig 1(b)). When the latent space is mapped through an affine transform to align latents on held-out trials to the true latent space, the Fourier domain GP-VAE does much better uncovering the true generative latent angle (Fig 1 (c), (d)).

## 3 THE MULTI-MODAL GAUSSIAN PROCESS VARIATIONAL AUTOENCODER

We now focus on extending the GP-VAE to model data of two modalities simultaneously. The examples we will emphasize here involve neural activity alongside some other experimental variable

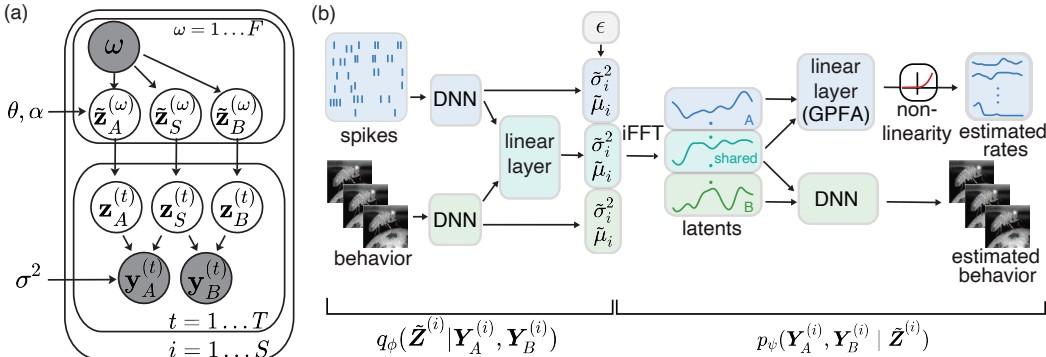

Figure 2: (a) Graphical model of the multimodal GP-VAE. A set of Fourier frequencies describe the Fourier representation of shared and independent latents across modality with a GP prior over each latent. Latents are transformed to the time domain and combined to generate data for each modality. (b) Schematic of the MM-GPVAE.

such as naturalistic movement or high-dimensional stimuli. The observations of our model are two distinct data modalities represented by matrices $\mathbf{y}_A^{(i)} \in I\!\!R^{N \times T}$ and $\mathbf{y}_B^{(i)} \in I\!\!R^{M \times T}$. Here, $t \in (1, 2 \ldots T)$ denotes time-bin indices, $i$ denotes trials, and $N$ and $M$ denote the dimension of the observations for each modality. We assume, as before, that all data are generated by smoothly varying low-dimensional latent variables that are now either independent to or shared between data modalities. As before, latents are initially parameterized in a pruned Fourier representation before being mapped to the time domain. The latents are then partitioned by a loadings matrix $W$ which linearly combines the shared latent representation with the independent latents.

$$\begin{bmatrix} \boldsymbol{x}_A \\ \boldsymbol{x}_B \end{bmatrix} = \begin{bmatrix} W_A & W_{S1} & 0 \\ 0 & W_{S2} & W_B \end{bmatrix} \begin{bmatrix} \boldsymbol{z}_A \\ \boldsymbol{z}_S \\ \boldsymbol{z}_B \end{bmatrix} + \mathbf{d} \tag{4}$$

Here, $\boldsymbol{z}_S$ refers to latents that are shared between modalities, while $\boldsymbol{z}_A$ and $\boldsymbol{z}_B$ denote modality-specific latent variables and $\mathbf{d}$ represents an additive offset. The outputs after this linear mixing result in modality-specific embeddings which we call $\mathbf{x}_A$ and $\mathbf{x}_B$. Here, modality $A$, the "neural modality", is passed through a pointwise nonlinearity $f$ to enforce non-negativity of Poisson rates, and modality $B$, the "behavioral modality", is passed through a deep neural network. The likelihood of the data given the embedding is:

$$p(\boldsymbol{y}_A | \boldsymbol{x}_A) \sim \mathcal{P}(f(\boldsymbol{x}_A)), \quad p(\boldsymbol{y}_B | \boldsymbol{x}_B) \sim \mathcal{N}(g_\psi(\boldsymbol{x}_B), \sigma_y^2 \boldsymbol{I}_N), \tag{5}$$

Where the function $g_\psi(\cdot)$ represents a decoder neural network for the behavioral data and we use the exponential nonlinearity for $f$. Learning is performed as before - the mean and variance of a mean-field Gaussian variational distribution is given by the data passed through a neural network with parameters $\phi$. Here, our lower bound is similar to that found in Casale et al. (2018) with an additional term for the neural data modality.

$$\text{ELBO} = \mathbb{E}_{\tilde{\boldsymbol{Z}} \sim q_\phi} \Big[ \overbrace{\sum_t \log(\mathcal{P}(\boldsymbol{y}_A | f(\boldsymbol{x}_A))}^{\text{Poisson Likelihood (Neural Rates)}} + \overbrace{\sum_t \log \mathcal{N}\big(\boldsymbol{y}_B \mid g_\psi(\boldsymbol{x}_B), \sigma_y^2 \boldsymbol{I}_N\big)}^{\text{Gaussian Likelihood (other modality)}}$$
$$+ \overbrace{\log p(\tilde{\boldsymbol{Z}} \mid \boldsymbol{T}, \boldsymbol{\theta})}^{\text{GP Prior}} \Big] + \overbrace{H(q_\phi)}^{\text{Entropy}} \tag{6}$$

A graphical depiction of the generative model of the MM-GPVAE is shown in Figure 2(a). The hyperparameters of the model are the latent-specific kernel parameters $\theta$ that provide the GP covariance structure as well as a constant $\alpha$ additive diagonal offset (fixed during inference). The entire schematic of the MM-GPVAE model, including the generative and variational distributions, is depicted in Figure 2(b).

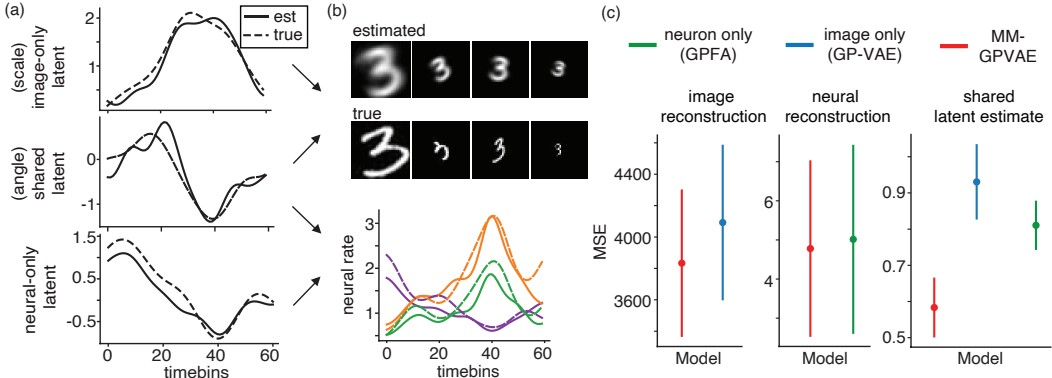

Figure 3: (a) True and estimated latents for the MM-GPVAE trained on simulated neural spiking data as well as a smoothly scaling and rotating MNIST digit. (b) Estimated neural rates on an example trial for 3 example neurons as well as 4 example reconstructed images of the MNIST digit at different angles and scales. (c) (*left*) Reconstruction accuracy from the image data trained on the images alone (GPVAE) compared to training with both modalities simultaneously (MM-GPVAE). (*middle*) Accuracy of estimated neural rates (*left*) trained on neural activity alone (Poisson - GPFA) compared to MM-GPVAE. (*right*). Accuracy of shared latent estimated from the MM-GPVAE compared to single-modality model variants. Error bars are standard error.

# 4 EXPERIMENTS

## 4.1 SIMULATED DATA

We first assess the performance of the MM-GPVAE model using a simulated dataset of two modalities: a smoothly rotating and scaling MNIST digit, and simultaneously accompanied Poisson spike counts from 100 neurons. A total of three latents were drawn from an underlying GP kernel to generate single-dimensional shared, neural, and image subspaces. There were a total of 300 trials, each trial consisting of 60 timebins. The data was split into 80% for training and 20% for testing. For this simulated example, one latent represents an interpretable modulation of the image as it directly affects the scaling of the MNIST digit. So as the latent values change along the trial the image scales smaller and larger continuously. Another latent, corresponding to the independent component of the neural modality, provides one component of the log rates of the Poisson spiking data. The final latent reflects the shared variability between both modalities. Here, the latent is again interpretable with respect to the image in that this latent corresponds to the angle of the smoothly rotating MNIST digit. This shared latent also linearly combines with the neural-only subspace to provide the log rates of the spiking Poisson population. The MM-GPVAE trained simultaneously on both images and spikes can successfully recover the three-dimensional latent structure across these two modalities. Figure 3(a) shows the underlying true latent in a held-out trial alongside the estimated latent structure extracted from the MM-GPVAE. As before, the held-out latent space is scaled to align with the true latent space as closely as possible, as the MM-GPVAE model latent space is invariant to scaling transform. In addition to accurately extracting the latent structure, the MM-GPVAE also has the ability to accurately reconstruct both neural rates and images across the time series from the latent space. Four example images and three example neural rates are shown in Figure 3(b). For more examples, see Figure 8 and 9.

The MM-GPVAE is also able to exploit information across modalities to better model the observed data. The left side of Figure 3(c) shows the mean squared error (MSE) across pixels for reconstructed images in held-out trials. By leveraging latent information from the spiking modality, the MM-GPVAE is able to reconstruct the images better than a single-modality GP-VAE trained on the images alone (Casale et al., 2018). This improvement is additionally dependent on the number of neurons analyzed (Fig 12). Similarly, the held-out neural rates are better estimated when the image data is included, though the effect here is more modest. Finally, we also show that the shared latent variable can be more accurately identified when data from both modalities is used (Fig 3(c)). Here we assess the accuracy of the shared latent estimate on test trials when we use data from each modality alone

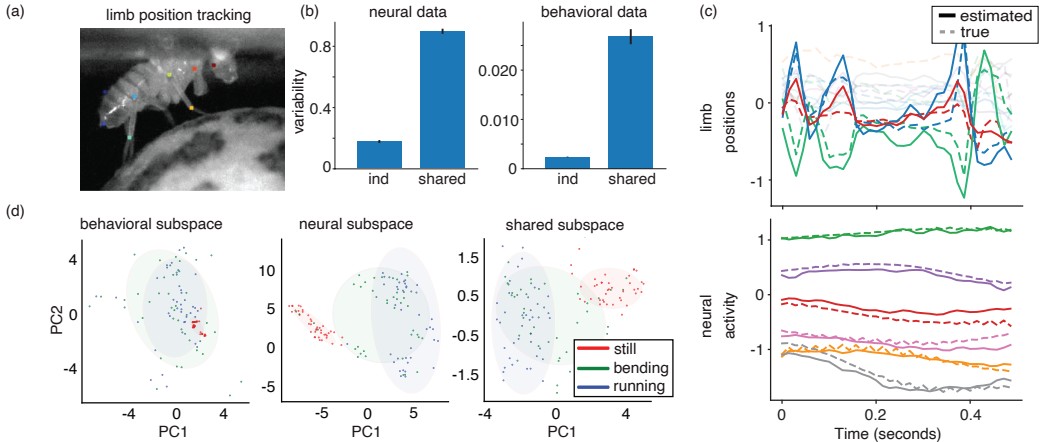

Figure 4: (a) Limb position tracking in *Drosophila* (b) (*top*) Contribution of the variability in the data across trials of the neural (*left*) and behavioral (*right*) modalities due to shared and independent subspaces (c) (*top*) Limb position estimates and true values and (*bottom*) six randomly selected calcium trace estimates and true values for a given trial. (d) Visualization of average latent value across time in the neural, behavioral and independent subspaces for 3 behavioral categories.

(akin to standard GP-VAE and GPFA) compared to both modalities simultaneously (MM-GPVAE). As expected, data from both modalities better identifies shared structure across the data types. This additionally does not depend on the structure of nature of the encoding distributions, (Fig 10), and is markedly improved by representing the MM-GPVAE with pruned Fourier frequencies (Fig 11).

## 4.2   APPLICATION TO FLY EXPERIMENTAL DATA

Next, we look to evaluate the MM-GPVAE on a real-world multi-modal dataset. Here, we consider a whole-brain calcium imaging from an adult, behaving *Drosophila* (Schaffer et al., 2021). We isolate 1000 calcium traces recorded using SCAPE microscopy (Voleti et al., 2019) from an animal while it performs a variety of distinct behaviors fixed on a spherical treadmill. These 1000 calcium traces are dispersed widely and uniformly across the central fly brain and were randomly picked from the dataset to use in our model. Alongside the neural measurements, eight 2-D limb positions are extracted from a recorded video using the software tool DeepLabCut (Mathis et al., 2018). These simultaneously recorded calcium traces and limb position measurements are split into 318 trials of 35 time-bins sampled at 70hz. Each trial has one of 5 corresponding behavioral labels (still, running, front grooming, back grooming and abdomen bending) determined via a semi-supervised approach from the tracked limb position measurements (Whiteway et al., 2021). Importantly, these behavioral labels are not used when fitting the MM-GPVAE. For additional information on how we isolate this behavioral data and neural traces, see the appendix. We consider a 7-dimensional latent behavioral subspace and 26-dimensional neural subspace, with 5 of these dimensions being shared across modalities. These choices were made through initial exploration of the model and examining cross-validated model performance (see appendix), though the results we show are robust to a wide range of dimensionality choices for each of the subspaces.

The MM-GPVAE is able to successfully reconstruct both behavioral trajectories and calcium traces in held-out trials for these data. The top of Figure 4(c) shows the true and decoded 16 limb position measurements with 3 highlighted for clarity. The bottom part of Figure 4(c) shows 6 randomly selected calcium traces alongside their model reconstructions. In each case, the MM-GPVAE can roughly capture the temporal trends in this multi-modal dataset. To analyze the shared and independent latent subspaces of these data we consider a 2d projection of each latent space, and plot the mean latent value in that subspace calculated each trial. We additionally color-code the trial according to the behavioral label. In these shared and independent latent subspaces, we find that the "still" behavioral state is well separated from many of the other behavioral conditions in the neural and shared subspaces. Here, we show two others (bending and running) as an illustration (Fig 4(d)). (For visualizing all 5 behaviors in the latent space, see Figure 19). Lastly, we illustrate how much each subspace contributes to the

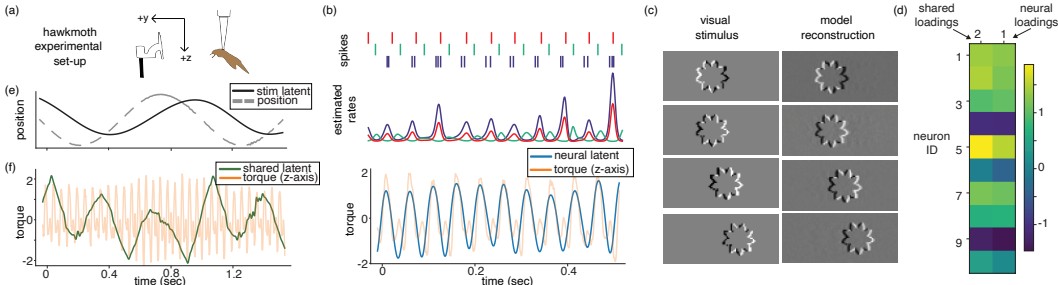

Figure 5: (a) Experimental set-up for *Manduca sexta*. Spikes from 10 muscle groups are recorded as an animal tracks a 1 Hz moving flower stimulus (b) *top*: Spikes from 3 example motor neurons *middle*: estimated Poisson rates *bottom*: The neural latent along torque measurement from the hawkmoth. The ~22Hz modulation reflects wing-flapping. (c) Visual stimulus and reconstructions from MM-GPVAE (d) Weights of hawkmoth spike decoder for neural-only and shared latents. (e) A one-dimensional latent from the visual-modality subspace closely tracks the stimulus position. (f) The shared latent between modalities plotted alongside the torque measurement.

overall variability of neural and behavioral data. Figure 4(b) shows the variance of the neural (left) and behavioral (right) data calculated for each trial where the reconstruction of the data is generated either from the shared subspace, or either of the independent subspaces. We find that the shared subspace for each modality contributes more to the overall variability in the data than either of the independent subspaces, suggesting a large fraction of shared variability between the data modalities.

## 4.3    APPLICATION TO MOTH EXPERIMENTAL DATA

Next, we evaluate the MM-GPVAE on a dataset of a hawkmoth (*Manduca sexta*) tracking a moving flower. The hawkmoth is an agile flier that is able to closely follow swiftly moving targets while hovering in midair, making it a model organism for the study of sensorimotor control (Sprayberry and Daniel, 2007). The modalities in our dataset consist of a time series of images of a visual stimulus temporally paired with electromyography (EMG) signals recorded during 20-second experimental trials. Each image is an event-based reconstruction (inspired by insect visual system) of a white floral target moving laterally during a trial as viewed in the hawkmoth's frame of reference (Sikandar et al., 2023) (Fig 5(a)). The floral target moves sinusoidally at 1 Hz against a black background. Each pixel in a video frame can occupy a polarity of -1, 0 or +1 based on whether its luminosity has respectively decreased, unchanged or increased as compared to the previous frame in the temporal sequence. Temporally paired with the floral stimulus, EMG recordings are precisely measured spike trains from the 10 major flight muscles of the hawkmoth responding to the floral target while flying on a tether and flapping at about 22 Hz. Altogether, the 10-muscle recordings form a near-complete motor program that controls its flight during target tracking response (Putney et al., 2019).

We fit the MM-GPVAE model to these data using a 2-dimensional latent space to describe the moving visual stimulus and a 2-dimensional latent space to describe the Poisson rates from the 10 motor units. Of these, one latent dimension is shared between modalities. Here, a small number of dimensions are used because only a few dimensions were needed to be able to accurately reconstruct the data of each modality (for more information about dimensionality selection, see appendix).

We find that the MM-GPVAE is able to accurately reconstruct both the neural rates (Fig 5(b)) and visual stimulus (Fig 5(c)) from its low dimensional latent space. We see that the neural rates are strongly modulated by the wing-flap oscillation at approximately 22 Hz. This is also seen in the neural latent space identified by the MM-GPVAE. At the bottom of Figure 5(b) we see that the 1-dimensional latent space closely aligns with the torque measurement along the z-axis of the moth motion, a measurement previously identified to be closely associated with wing beating (Putney et al., 2019). The MM-GPVAE is also able to reconstruct the visual stimulus data, as seen in 5(c). When viewing the latent space corresponding to the visual stimulus modality, we find that the value of the smoothly evolving latent variable closely tracks the position of the stimulus in x dimension (the only dimension along which it moves, see Fig 5(e)). Finally, the shared latent dimension shows some modulation with wing-flapping but exhibits temporal structure not obviously related to the flapping

frequency or stimulus motion. This may correspond to a longer-timescale motor behavior such as the variation in how strongly the hawkmoth is responding to track the stimulus, which could depend on the degree of its attention to the stimulus and its state of motivation. Importantly, we find that including a shared dimension better reconstructs the data than not including one, suggesting that there is a component of neural dynamics shared with stimulus dynamics. Specifically, we find that if we remove the shared latent we see approximately a 70% reduction the reconstruction accuracy of the image modality and about a 5% reduction in the reconstruction of the rates (Fig 21). Because the neural rates have a strong wing-flapping modulation, we expect the slower-moving dynamics to account for only a small amount of the variability here. Lastly, to demonstrate the interpretability advantage of linear decoder layer used for the neural likelihood, we visualize how each of the 10 motor neurons is loaded onto either the neural-only dynamics or the dynamics shared with the moving stimulus Figure 5(d). For a closer look a individual neurons modulations by shared and independent latent components, see Figure 21(b).

## 5 CONCLUSION

In this work, we have introduced the multi-modal Gaussian Process variational autoencoder (MM-GPVAE) to identify temporally evolving latent variables for jointly recorded neural activity and behavioral or stimulus measurements. We first parameterize the model in the Fourier domain to better extract identifiable temporal structure from high-dimensional time-series data. We then combine a linear decoder for neural data with a deep neural network decoder for behavioral or stimuli data to enhance the applicability of the MM-GPVAE across experimental settings while retaining the ability to identify neural tuning with respect to shared and independent temporally evolving latents. We show that our multi-modal GPVAE can accurately recover latent structure across data modalities, which we demonstrate using a smoothly rotating and scaling MNIST digit alongside simulated neural spike trains. We then show that the MM-GPVAE can be flexibly adapted to multiple real-world multi-modal settings. First, we analyze calcium imaging traces and tracked limb positions of fruit flies (*Drosophila*) exhibiting various distinct behaviors. Then, we apply the model to spike trains from hawkmoth flight muscles as well as the oscillating visual stimulus the animal tracks during flight.

**Comparison to multi-modal approaches in neuroscience:** Our work contrasts in important ways with two similar recently-developed multi-modal latent variable models for time series data in neuroscience. One closely related model, *Targeted Neural Dynamical Modeling* (TNDM) (Hurwitz et al., 2021), was specifically designed to non-linearly separate the neural latent representation into behaviorally relevant and irrelevant subspaces using a latent space governed by a recurrent neural network (RNN). However, this work was evaluated on a dataset with a relatively simple behavioral paradigm, and fails to accurately model the behavioral modality when it is higher dimensional or more complex (Fig 6). Another related multi-modal time-series neuroscience model, *Preferential Subspace Identification* (PSID) (Sani et al., 2021), linearly separates the neural latent representation into behaviorally relevant and irrelevant subspaces. However, PSID may be limited in that uses latent dynamics are governed by a linear state-space model, and, similar to TNDM, is restricted in capturing complex behavioral modalities well due a linear decoder to behavioral data (Fig 6). For a more thorough evaluation and discussion to these approaches, see the appendix.

**Choice of prior:** In this work we focus on the use of GP priors to describe latent dynamics, though other temporal dependencies are of course possible. Though TNDM and PSID have certain limitations when compared with the MM-GPVAE (Fig 6), these are not necessarily due to their latent temporal prescriptions. These models could be modified with more flexible decoding schemes to better isolate the role of the different latent dynamics in characterizing shared and independent subspaces compared to the MM-GPVAE. In addition, other choices such as switching linear or non-linear latent dynamics models are powerful in single modality settings (Hernandez et al., 2018; Linderman et al., 2017; Glaser et al., 2020) and could also be adapted to multi-modal settings. Our choice of a GP prior specifically is well suited to experimental paradigms where the latent dynamics are hypothesized to be smooth. Other approaches might be more appropriate when this smoothness assumption is unlikely to hold. Thus, development of multi-modal versions of different latent dynamical models in neuroscience and evaluating their strengths and limitations alongside the MM-GPVAE remains an important avenue of future work in neuroscience.

## 6 REPRODUCIBILITY STATEMENT

All models in this manuscript were trained end-to-end in PyTorch using the ADAM optimizer. Training was done on a Macbook Pro with Apple M1 max chip and all evaluations took less than an hour to fit. All encoder and decoder neural networks were standard feedforward neural networks with ELU activation functions. Additional details on neural network architectures for the experiments can be found in the appendix. An implementation of MM-GPVAE can be found at: GitHub Repository for MM-GPVAE.

## 7 ETHICS STATEMENT

Our work provides a general model for exploratory data analysis for multi-modal time-series data. Though we develop the model with neuroscience experiments in mind, in principle the model could be adapted to other settings where the assumption of smoothly evolving latent dynamics holds across distinct data modalities. We do not anticipate any potential negative societal impacts of our work.

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

APPENDIX

CONTRASTING THE MM-GPVAE WITH EXISTING APPROACHES

We evaluate our MM-GPVAE model alongside both Targeted Neural Dynamical Modeling (TNDM) (Hurwitz et al., 2021) and Preferential Subspace Identification (PSID) (Sani et al., 2021) using our synthetic data of a rotating and scaling MNIST digit '3' and corresponding Poisson neural rates. In Figure 6, we show the reconstruction of both the behavioral modality, the MNIST digit '3', and Poisson neural rates for each model. In addition to these, we also provide the error of the reconstruction performance on both data modalities on held-out trials. We find that the MM-GPVAE is able to better reconstruct data from both modalities in this simulated example than the competing approaches.

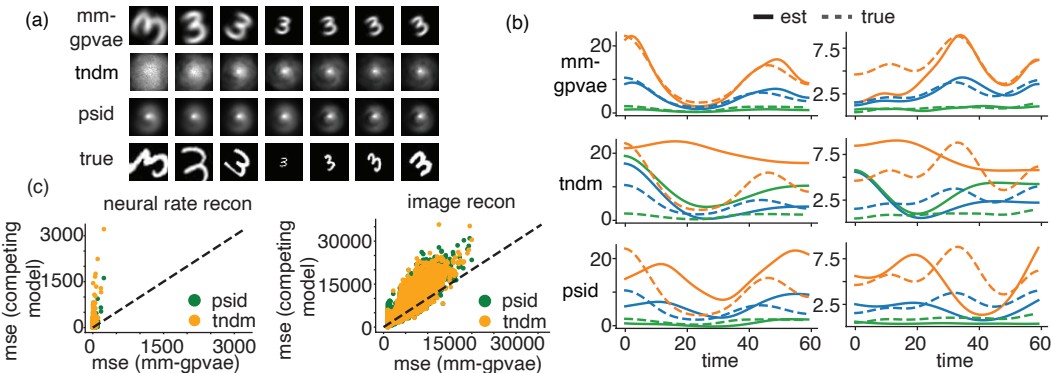

Figure 6: (a) Reconstruction of the scaling/rotating MNIST digit '3' with MM-GPVAE, TNDM and PSID. (b) Reconstruction of neural rates with MM-GPVAE, TNDM, and PSID. (c) MSE for neural rate reconstruction (left), and MSE for image reconstruction (right). Here, each dot indicates one trial mse from MM-GPVAE vs a competing model. The majority of the trials errors fall above the unity line for both models, indicating overall better reconstruction with MM-GPVAE.

Next, to compare these models in a real-world setting, we implement PSID, TNDM, and MM-GPVAE on data from a simpler neuroscience experiment where all models are capable of recovering both behavioral trajectories as well as neural rates. We focus on a simpler behavioral task because these competing approaches are not well-suited to reconstruct complex behavioral modalities due to a lack of neural network decoder. We use the primate reaching data from Hurwitz et al. (2021) for these evaluations. Here, a linear decoder as well as our deep neural network can reconstruct both modalities well. The visualizations of the latent spaces for these data can be seen in Figure 7. While both TNDM and PSID can provide only two subspaces in which only one can separate the 8 reaching positions (relevant subspace), MM-GPVAE can provide visualization for 3 subspaces, 2 modality specific, and 1 shared, and we can see a clear separation with both the behavioral-only subspace as well as shared subspace.

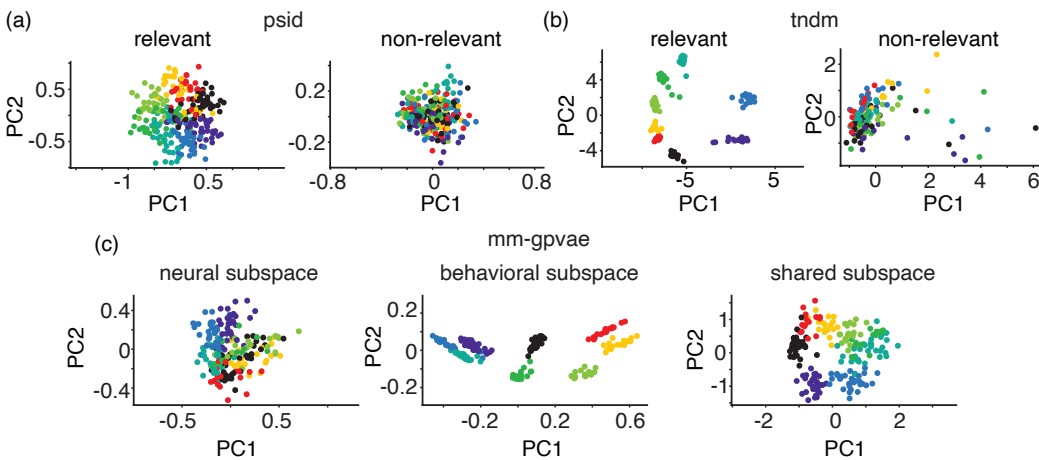

Figure 7: (a) Separation of all 8 reaching directions in the relevant subspace using PSID. (b) Separation of all 8 reaching directions in the relevant subspace using TNDM. (c) Separation of all 8 reaching directions in behavior-only and shared subspaces. Dots here indicate the mean latent value across the entire trial. The neural subspace shows no behavioral separation in the latent-space whereas the behavioral and shared subspaces show strong behavioral separation. This result closely parallels with (Hurwitz et al., 2021; Sani et al., 2021), which isolates behaviorally relevant and irrelevant neural subspaces. However, in contrast to (Hurwitz et al., 2021; Sani et al., 2021) the MM-GPVAE isolates a distinct shared subspace as well as both neural and behavioral independent subspaces from a raw unsupervised partitioning of both the behavioral and neural data.

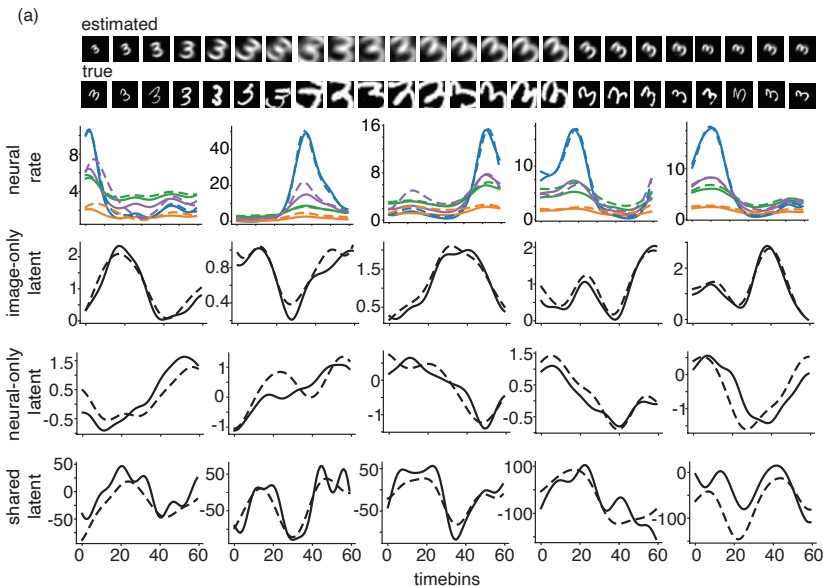

Figure 8: (a) Additional estimated images and neural rate trajectories of the MNIST digit '3' as well as independent and shared latent estimates.

## ADDITIONAL EVALUATIONS OF THE MM-GPVAE ON SIMULATED DATA

To provide a more complete picture of the ability of the MM-GPVAE to both reconstruct simulated behavioral and neural data, and to accurately recover the true underlying latent trajectories, we show additional performance evaluations here in a simulated setting. Figure 8 shows an example of 24 reconstructed 3s from the evaluation shown in Figure 3 of the manuscript. Again, here we reconstruct a scaling and rotating MNIST digit '3' alongside 100 neural spike trains, where one latent dimension is shared across modalities. Figure 8 also shows latent reconstructions and 5 example neural rates in 5 held out trials. Using the same set-up, we additionally run the MM-GPVAE with MNIST digit '2', and show an example of 24 reconstructed images as well as 3 example neural rates (of 100 neurons) on 5 held-out trials in Figure 9.

In addition to these, we extend Figure 3(a) from our manuscript with all the latent trajectories for our simulated data comparisons. Because in our unimodal (GPFA and GP-VAE) comparisons, there is ambiguity as to which latent is "shared", we simply select the latent trajectory which most closely matches the true underlying latent variable for each comparison. Panels (b) and (c) in Figure 10 show the reconstruction mean squared error of the MM-GPVAE compared to the unimodal models GP-VAE and GPFA. Here, we include a version of each of the unimodal models with two different encoder representations, one which represents what the GPFA and MM-GPVAE would typically have access to - simply the data of their own modality (i.e. neural data for GPFA, image/behavioral data for GP-VAE) - and a second where the encoding distribution is a function of data from both modalities, matching the encoder exactly with that of the MM-GVPAE. In each of these cases, we find that the encoding representation has little effect on model performance. We visualize both the average mean-squared error on held-out data (c) as well as a per-trial mean-squared error scatter-plot (b) to provide a better sense of performance across all models and encoders. We note though that the extent to which MM-GPVAE more accurately identifies latent structure than the unimodal variants is sensitive to the nature of the simulated data and can vary across model fits. For (b) and (c) specifically, we average model performance across 4 random seeds to report overall reconstruction accuracy. We also found through model exploration that GPFA specifically often does comparatively well in identifying latent structure when neural rates are high or have large variance. This is likely true because identifying latents within a linear map is trivial under the standard GPFA model, and so latent accuracy can be very good here if the neural data is sufficiently robust. This means that the metrics reported for the identification of latent structure in Figures 10 and 3 are somewhat dependent on the statistics of the neural rates used to generate and fit the model.

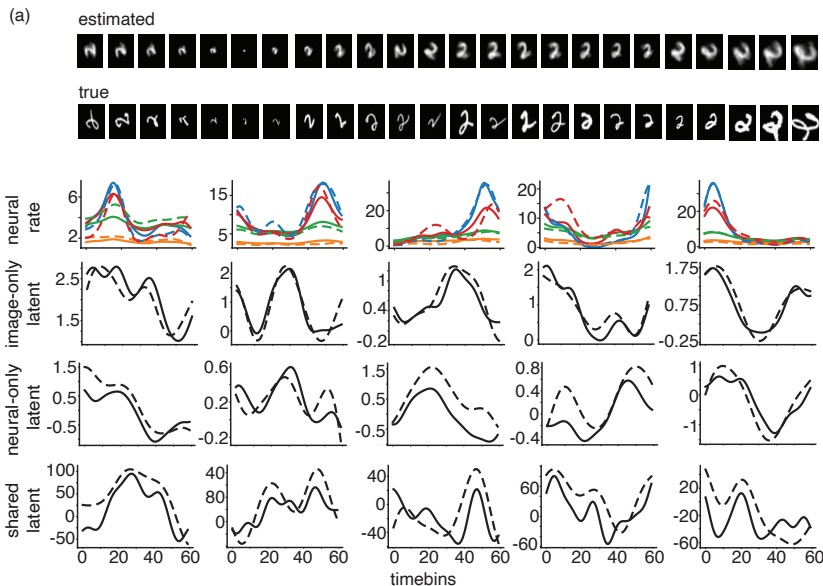

Figure 9: (a) Example of MNIST digit '2' with reconstructed digits and neural rates as well as independent and shared latent estimation.

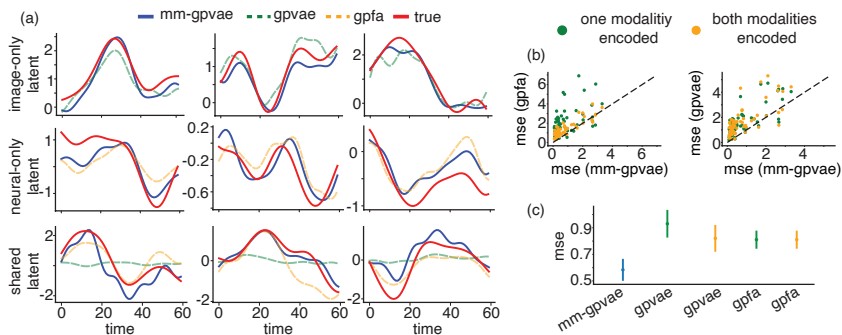

Figure 10: (a) Extension of Fig 3(a) from the manuscript with best latent trajectories, showing the best matched latent from GPFA and GP-VAE to the true shared and independent latents. (b) Identification acccuracy of the best performing latents estimating the shared latent for GPFA (left) and GP-VAE (right). Each dot indicates a trial. (c) Average MSE for recovery of shared latent for the best possible latents from GP-VAE, GP-VAE (encoding both modalities) GPFA, and GPFA (encoding both modalities).

ASSESSING THE IMPORTANCE OF PRUNING THE FOURIER FREQUENCIES

Figure 11 demonstrates the necessity of pruning the Fourier frequencies when evaluating the MM-GPVAE on our simulated data. Here, we present standard and pruned Fourier MM-GPVAE model fits (Fig 11(a)) and show their ability in reconstructing the true generative smooth latent across each latent subspace (Fig 11(b)). Without constraining the number of Fourier features for these data, the latent identification fails, akin to what is seen when we ignore the Fourier representation entirely. We note here that this feature is a function of the length of the trial, and the failure is not as stark for shorter-length trials. For these trials, our pruned condition the length scale of the GP to a value of 10 (i.e. the hyperparameter $\ell$ can go no smaller than 10).

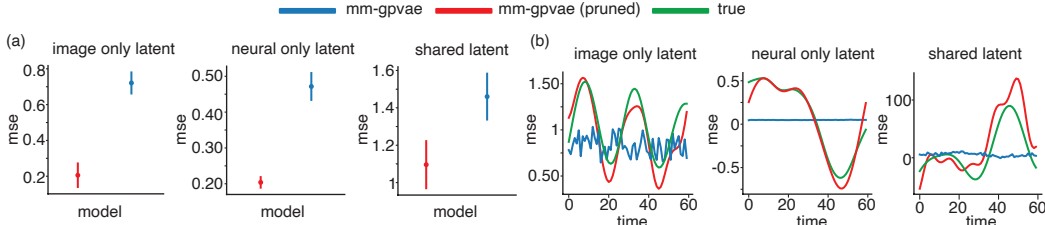

Figure 11: (a) Accuracy of identification of all three latents for the pruned and standard version of the Fourier MM-GPVAE. (b) Example latent trajectories for each subspace. The time-domain formulation fails to identify the true underlying latents accurately, similar to what is seen in Fig 1.

VARYING THE NUMBER OF NEURONS

To demonstrate how the recovered latent of the shared and independent subspaces might trade-off with the amount of data a practitioner may have for a given modality, we consider fits of our MM-GPVAE in our simulated setting with a varying number of neurons. As expected, increasing the number of neurons increases the ability of the MM-GPVAE to identify the neural latent, as well as its ability to accurately reconstruct neural data. Importantly, because structure is shared across modalities, increasing the number of neurons also increases the ability of the MM-GPVAE to identify shared latent structure, as well as shows some improvement in the MM-GPVAE to generate image data. Increasing the number of neurons has no effect on identifying the image latent structure. (See Fig 12)

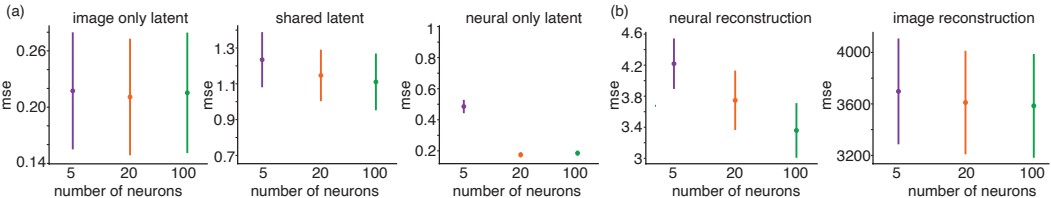

Figure 12: (a) Latent identification accuracy as a varying number of neural rates are available to the model. (b) image and rate reconstruction accuracy as a varying number of neural rates are available to the model.

ROBUSTNESS TO MEASUREMENT ERROR

To assess how the MM-GPVAE might perform in the presence of noisy or imperfect behavioral measurements, we consider a simulated experiment where a randomly selected 20% of the images are greyed on either the top or bottom half. We fit the MM-GPVAE jointly to the image data of this form as well as uncorrupted neural rates of the same form as in the main manuscript. We find that the MM-GPVAE is able to exploit smoothness, rendering well-generated images even during these occluded trials (Fig 13(a)). As expected, we find that a standard MM-VAE (without a GP prior over time) is not able to reconstruct the true image as well, but does a better job reconstructing the occluded image, as it treats each frame independently (Fig 13(b)). We also show that this occlusion procedure has no bearing on the ability of the MM-GPVAE or MM-VAE to reconstruct the neural rates (Fig 13(c)).

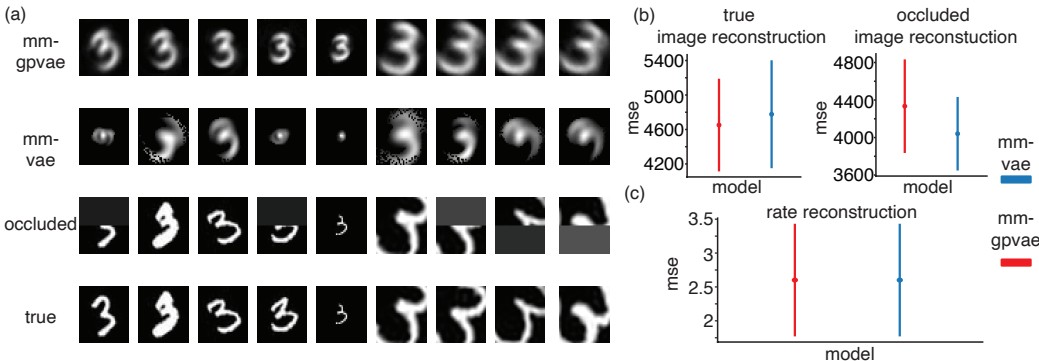

Figure 13: (a) Image reconstructions of MM-VAE and MM-GPVAE trained on occluded threes alongside the true and occluded threes. (b) (left) MSE of MM-GPVAE and MM-VAE for the true, not occluded data. (right) MSE of MM-GPVAE and MM-VAE for the occluded data. (c) MSE of rate reconstructions.

## NEURAL NET ARCHITECTURES

For the MM-GPVAE, both the simulated and real-world multi-modal evaluations used similar neural net architectures. However, there were some modifications of the nodes/layers that were unique to each evaluation. This was necessary as the latent dimensionality was different across our different evaluations, and certain behavioral reconstructions, especially those that were higher-dimensional, required richer neural network parameterizations. The schematic for the MM-GPVAE neural network architecture in our simulated example can be found in Figure 14, and the schematics for the neural network architectures for the real-world multi-modal datasets can be found in Figure 15 (fly) and Figure 16 (hawkmoth). For our evaluation on the dataset used in (Hurwitz et al., 2021), whose results are above, the architecture can be found in Figure 17. Note that for all experiments, each modality is encoded to its own set of variational means and variances (transformed into the Fourier domain). The encoded means and variances representing the shared latents are then summed to give the encoded shared latents means and variances. Across all evaluations, we parameterized our latents in the Fourier domain and converted back to the time domain before decoding.

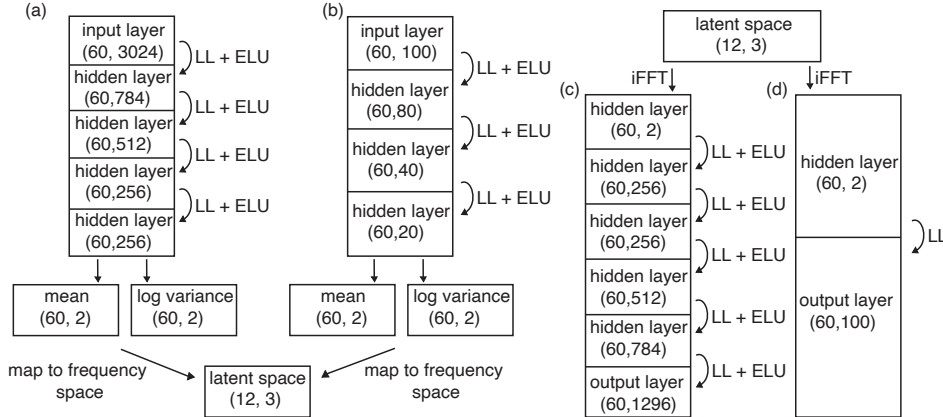

Figure 14: MM-GPVAE architecture for simulated data. (a) Encoder network of the MNIST digit. (b) Encoder network of the neural information. (c) Decoder network of the MNIST digit. (d) Decoder network of the neural information.

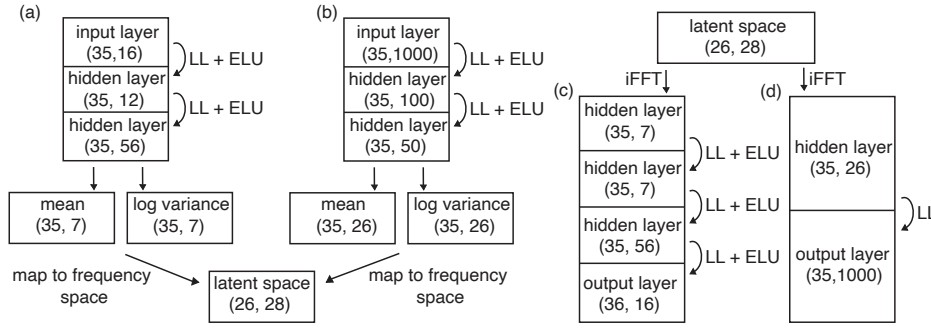

Figure 15: Neural network architecture for evaluation on fly dataset. (a) Encoder network for behavior. (b) Encoder network for neural data. (c) Decoder network for behavior. (d) Decoder network for neural data.

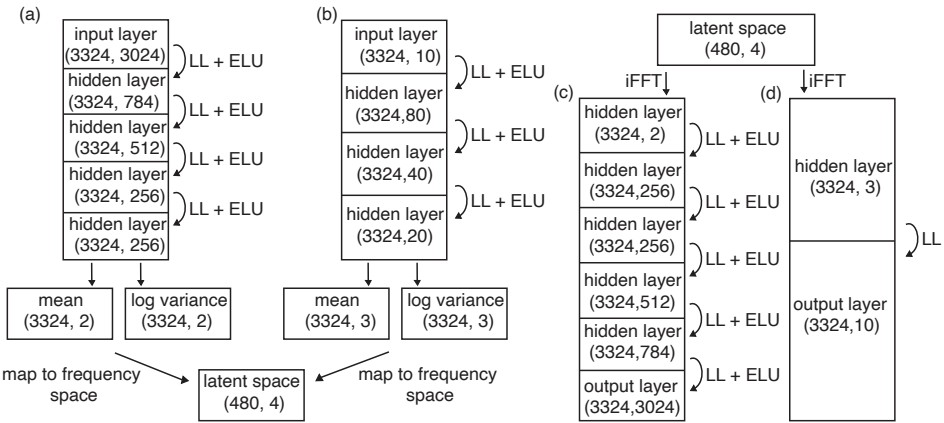

Figure 16: Neural network architecture for evaluation on hawkmoth dataset. (a) Encoder network for behavior. (b) Encoder network for neural data. (c) Decoder network for behavior. (d) Decoder network for neural data.

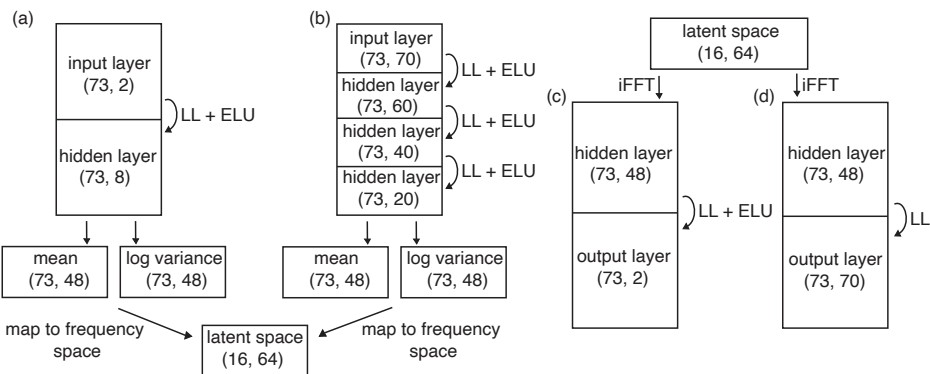

Figure 17: Neural network architecture for Monkey reaching data, evaluated above. (a) Encoder network for the behavior. (b) Encoder network for the neural activity. (c) Decoder network for the behavior. (d) Decoder network for the neural activity.

ADDITIONAL INFORMATION ON FLY EXPERIMENTAL DATA

*Pre-processing:* We isolated 1000 raw calcium traces from (Schaffer et al., 2021) for use with our model for the evaluations on the *Drosophila* dataset. For this set-up, we consider a variant of the MM-GPVAE where we have removed the non-linearity for the neural data modality, and instead consider Gaussian observations with an additional parameter controlling the observation variance for the fluorescence traces (akin to the original formulation of GPFA (Yu et al., 2009)). This dataset also contained x,y positions from 8 tracked limbs positions (Mathis et al., 2018). The data came as one continuous recording, and at every time point there was a behavioral categorization probability for one of 6 distinct behaviors (undefined, still, running, front grooming, back grooming, abdomen bending) determined via the algorithm outlined in (Whiteway et al., 2021). To split this continuous recording into trials, we found segments of the recording where, for 35 continuous samples, a single behavior was estimated at a $\geq \%60$ probability. This generated 318 total trials where each trial was of one of 5 possible behaviors. There was no section of the recording where there was an 'undefined' behavior for 35 continuous samples, so there are no undefined behavioral trials in our analysis.

*Additional evaluation:* Though we show the time-course of the behavioral reconstruction in the main manuscript, we show a 2-d reconstruction on 2 example trials in Figure 18. Here we plot true x,y positions for 2 trails for each of the 8 limb positions alongside our model's reconstruction. We can see here that our model also captures the spatial information in the limb-position data. We also show for comparison the 2-d depiction of the shared and independent latent representation of all trials in the dataset (Figure 19), with all five of the behaviors labelled. You can see here that the "still" behavior separates well from all other behaviors in the shared and neural subspaces.

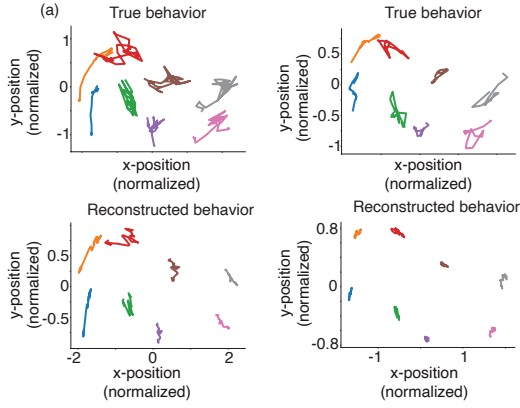

Figure 18: (a) Reconstruction of 8 fly limb positions in 2 held-out trials. Here, we see the MM-GPVAE is able to reconstruct the spatial information of the behavioral modality in the 8 tracked limb positions.

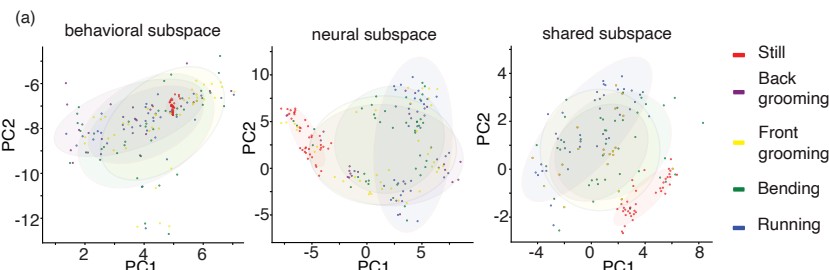

Figure 19: (a) Separation of all 5 fly behaviors in the behavior-only, neural-only, and shared subspaces. Similar to what is seen in the main manuscript, the 'still' behavior is well separated from trials with other behavioral labels in the neural and shared latent subspaces.

ADDITIONAL INFORMATION ON HAWKMOTH EXPERIMENTAL DATA

*Pre-processing:* For our hawkmoth data, the original synthetic visual stimuli were sampled at 125 Hz and the neural and torque recordings were sampled at 10K Hz (Sprayberry and Daniel, 2007; Sikandar et al., 2023; Putney et al., 2019). To prepare these data for evaluation with the MM-GPVAE, we first downsampled neural information and torque measurements to 1K Hz by binning the spike counts and averaging the torque measurements at this temporal resolution. To align these measurements with the visual stimuli, we upsampled the images 8-fold. The entire recording was 20 seconds long and was split into 6 evenly-divided trials. Here, it is worth noting that the trials are much longer ($\sim 3300$ timebins) than in the other experiments in this work.

The visual stimuli contained 3024 pixels and there were ten total hawkmoth motor neurons. We set the neural-independent subspace to 1-dimensional, the images-independent subspace to 1-dimensional, and an additional 1 dimension for the shared subspace. To encourage slow-evolving smooth latents in the shared and image subspaces, and faster-evolving neural latents, we initialized the length scale parameters for each latent dimension to different values. The length scale was set to 10 for the neural latents, 150 for the shared latent, and 300 for the image latent. This biased the model fits to capture slower dynamics in the image subspace and the faster dynamics in the neural subspace.

*Additional evaluation:* The neural rates are well-captured by the MM-GPVAE for all 10 hawkmoth motor neurons, which each show strong periodicity. Figure 20 shows the strong fast-oscillation spike rates captured by the MM-GPVAE for all ten hawkmoth neurons alongside the recorded spikes for a 1.5 second period. Similar periodicity exists across the entire 20 second recording.

To emphasize the added interpretability we get from the linear layer in the neural likelihood and the role of the shared latent in generating the data from each modality, in Figure 21 we show the reconstructions for both modalities using one or no shared latent variables (Fig 21 (a)) as well as 2 example neurons' tuning to the neural and shared latent. Here we can see that neuron 5 is more positively modulated by the shared latent while neuron 9 is negatively modulated by the shared latent.

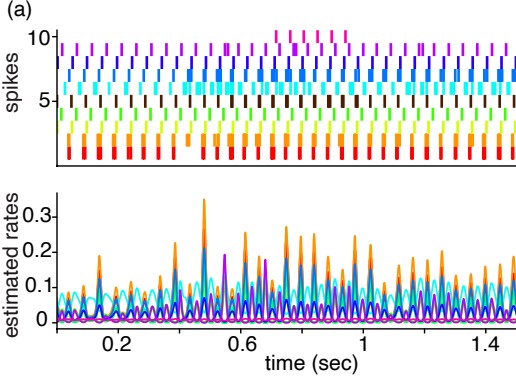

Figure 20: (a) True spikes and estimated rates of all 10 hawkmoth neurons over 1500 time points.

ADDITIONAL MODEL FITTING INFORMATION

All data had a 80-20 split for training and testing respectively. The Fourier frequency pruning was set to the minimum length scale of 10, 10, 3, and 16 for GP-VAE (simulated), MM-GPVAE (simulated), MM-GPVAE (fly), and MM-GPVAE (moth) respectively. GP length scales parameters were initialized to a value of 30 for all except for the hawkmoth evaluations (where initial values are indicated above), and jointly optimized with the ELBO. The covariance parameter $\alpha$ was set at a fixed value of 1e-2, 1e-2, 1e-3, and 1e-4 for GP-VAE (simulated), MM-GPVAE (simulated), MM-GPVAE (fly), and MM-GPVAE (moth) respectively. We additionally initialized the offsets $\mathbf{d}$ of the neural modality to the average log-rate of the neural data.

**Latent dimensionality selection**: In the case of the simulated examples in this work, the dimensionality was always set to the true generative dimensionality. For the real neural dataset examples in the main paper, the dimensionality was chosen using a distinct approach for each modality. For the

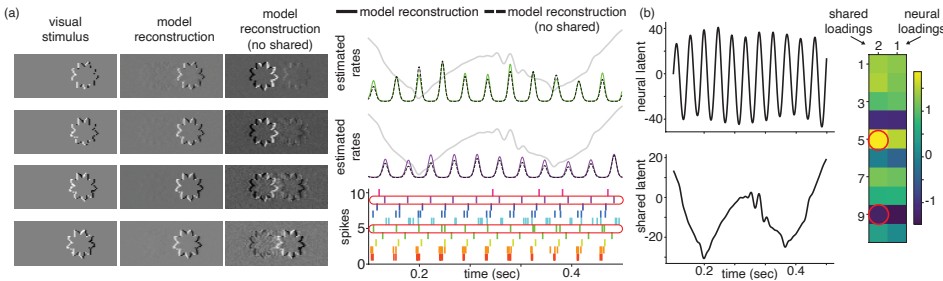

Figure 21: (a) Stimulus reconstructions using one shared latent vs no shared latent variables (left). Neural rate reconstructions using one shared latent vs no shared latent (right). (b) Fast moving neural latent (top) along with slow moving shared latent (bottom). The loadings matrix (right) display the tuning of each neuron to our latents, here we highlight neuron 5 which is positively modulated by the shared latent and neuron 9 which is negatively modulated modulated by the shared latent.

neural dataset, we use PCA and select the number of dimensions that capture greater than 97% of the variance. For the other modality (limb position or visual image) the dimensionality was chosen by increasing the number of dimensions in a unimodal Fourier GP-VAE until cross-validated performance did not show significant improvement. Then, we identify the appropriate number of shared dimensions by fitting the full MM-GPVAE model to both modalities and systematically increasing the number of dimensions shared between datasets until predictive performance no longer improved. Across both datasets, we always saw that adding some shared dimensions increased overall predictive performance compared to none, suggesting shared structure across the data modalities for these experiments.

We also note that the dimensionality selection procedure was used to approximate shared and independent dimensionality as best as possible, but across CV folds the variance in cross validated log-likelihood was high. However, small changes to the number of shared and independent dimensions did not change latent representations or decoding performance much in our models.

For the evaluation where we compare MM-GPVAE to TNDM and PSID, we choose approximately the same dimensionality as in the original TNDM paper (Hurwitz et al. (2021)) so readers can easily compare fits across models.

### CHOICE OF GAUSSIAN OBSERVATION VARIANCE INITIALIZATION, $\sigma^2$

We found that how we initially set the value of $\sigma^2$ could effect the performance of the MM-GPVAE, especially regarding the reconstruction of the behavioral data. Although we learned the $\sigma^2$ value jointly with the other parameters during optimization, the value of $\sigma^2$ tended to vary minimally from its initial value. Therefore, our setting of $\sigma^2$ tended to act as a reconstruction penalty, balancing the ability of the MM-GPVAE to prefer to reconstruct either the behavioral or neural data. Such a scaling term in the ELBO reconstruction has been seen in other models (Hurwitz et al., 2021), and in the original GP-VAE this parameter was chosen carefully through a specific cross-validation approach (Casale et al., 2018). Here, we simply set $\sigma^2$ to a value proportional to the dimensionality of the behavioral modality, which tended to balance the neural and behavioral terms in the ELBO, and generate good reconstructions for each modality. The initial values for $\sigma^2$ were 1000, 100, 1e-6, and 1 for GP-VAE (simulated), MM-GPVAE (simulated), MM-GPVAE (fly), and MM-GPVAE (moth) respectively.

### DERIVATION OF THE EVIDENCE LOWER BOUND (ELBO)

Here we show the derivation of the evidence lower bound used for the MM-GPVAE. For clarity, we will start by deriving the ELBO for the MM-GPVAE in the time domain, and ignore the Fourier representation in this derivation. The derivation trivially applies to the Fourier representation as well, as the mapping between the two domains is linear. We set the GP prior parameters $\boldsymbol{\theta} = \{\alpha, \theta\}$, $\boldsymbol{W}$ to be the loadings matrix and offsets from equation 4 in the manuscript, $\boldsymbol{T}$ to be the observed timepoints,

and $\boldsymbol{Z}$ to be the collection of all the shared and independent latents $\boldsymbol{Z} = \{\mathbf{z_a}, \mathbf{z_s}, \mathbf{z_b}\}$.

$$
\begin{aligned}
\log p\left(\boldsymbol{Y}^A, \boldsymbol{Y}^B \mid \boldsymbol{T}, \boldsymbol{\phi}, \boldsymbol{W}, \sigma^2, \boldsymbol{\theta}\right) = \\
\log \int \frac{p\left(\boldsymbol{Y}^A, \boldsymbol{Y}^B \mid \boldsymbol{Z}, \boldsymbol{\phi}, \boldsymbol{W}, \sigma^2\right) p(\boldsymbol{Z} \mid \boldsymbol{T}, \boldsymbol{\theta})}{q_{\boldsymbol{\psi}}(\boldsymbol{Z} \mid \boldsymbol{Y}^A, \boldsymbol{Y}^B)} q_{\boldsymbol{\psi}}(\boldsymbol{Z} \mid \boldsymbol{Y}^A, \boldsymbol{Y}^B) d\boldsymbol{Z} \\
\geq \int \log \left( \frac{p\left(\boldsymbol{Y}^A, \boldsymbol{Y}^B \mid \boldsymbol{Z}, \boldsymbol{\phi}, \boldsymbol{W}, \sigma^2\right) p(\boldsymbol{Z} \mid \boldsymbol{T}, \boldsymbol{\theta})}{q_{\boldsymbol{\psi}}(\boldsymbol{Z} \mid \boldsymbol{Y}^A, \boldsymbol{Y}^B)} \right) q_{\boldsymbol{\psi}}(\boldsymbol{Z} \mid \boldsymbol{Y}^A, \boldsymbol{Y}^B) d\boldsymbol{Z} \\
= \mathbb{E}_{\boldsymbol{Z} \sim q_{\psi}} \left[ \log p(\boldsymbol{Y}^B \mid \boldsymbol{\phi}, \sigma^2, \mathbf{z_s}, \mathbf{z_b}) + \log p(\boldsymbol{Y}^A \mid \boldsymbol{W}, \mathbf{z_s}, \mathbf{z_a}) + \log p(\boldsymbol{Z} \mid \boldsymbol{T}, \boldsymbol{\theta}) \right] \\
- \int \log q_{\boldsymbol{\psi}}(\boldsymbol{Z} \mid \boldsymbol{Y}) q_{\boldsymbol{\psi}}(\boldsymbol{Z} \mid \boldsymbol{Y}) d\boldsymbol{Z} \\
= \mathbb{E}_{\boldsymbol{Z} \sim q_{\psi}} \left[ \sum_t \log \mathcal{N}\left(\boldsymbol{y}_B \mid g_{\boldsymbol{\phi}}\left(\boldsymbol{x}_B\right), \sigma^2 \boldsymbol{I}_N\right) + \sum_t \log(\mathcal{P}(\boldsymbol{y}_A | f(\boldsymbol{x}_A)) \right. \\
\left. + \log p(\boldsymbol{Z} \mid \boldsymbol{T}, \boldsymbol{\theta}) \right] + H(q_{\psi})
\end{aligned}
$$

The Fourier domain representation of the ELBO only requires sampling over variational parameters in the Fourier space, but only changes the expression of the GP prior term $p(\boldsymbol{Z})$.

*GP prior*: The expectation of the GP prior term can be expressed in the Fourier domain as:

$$
\begin{aligned}
\mathbb{E}_{\tilde{\boldsymbol{Z}} \sim q_{\psi}} \left[ p(\tilde{\boldsymbol{Z}} \mid \boldsymbol{\theta}, \boldsymbol{\omega}) \right] &= \sum_{p, \omega} \mathbb{E}_{\tilde{\boldsymbol{Z}} \sim q_{\psi}} \left[ \log \mathcal{N}\left( \tilde{z}_{p, \omega} | 0, [\tilde{\boldsymbol{K}}_p]_{\omega, \omega} \right) \right] \\
&= \frac{1}{2} \sum_{p, \omega} \mathbb{E}_{\tilde{\boldsymbol{Z}} \sim q_{\psi}} \left[ \log([\tilde{\boldsymbol{K}}_p]_{\omega, \omega} + \alpha) + \frac{\tilde{z}_{p, \omega}^2}{([\tilde{\boldsymbol{K}}_p]_{\omega, \omega} + \alpha)} \right] \\
&= \frac{1}{2} \sum_{p, \omega} \log([\tilde{\boldsymbol{K}}_p]_{\omega, \omega} + \alpha) + \frac{\tilde{\sigma}_{p, \omega}^2(\boldsymbol{Y}) + \tilde{\mu}_{p, \omega}^2(\boldsymbol{Y})}{([\tilde{\boldsymbol{K}}_p]_{\omega, \omega} + \alpha)},
\end{aligned}
$$

where the double sum is due to the variational distribution $q$ being a mean field Gaussian, and $p$ here indexes latents and $\omega$ indexes Fourier frequencies.

*Neural likelihood*: Since the estimated log-rates of the neural data is a linear transform of the shared and neural latent variables, we can also evaluate the expectation of the neural-modality likelihood term in closed form. Recall that

$$
\boldsymbol{X} = W\boldsymbol{Z} = W\tilde{\boldsymbol{Z}}\boldsymbol{B}^\top,
$$

where $\boldsymbol{X}$ is the matrix of embeddings, and $\tilde{\boldsymbol{Z}}$ is the $P \times \mathcal{F}$ matrix of Fourier-domain latent variables. We may therefore write the embedding for measurement $i$ at time $t$ as

$$
x_{i,t} = \boldsymbol{w}_i^\top \tilde{\boldsymbol{Z}} \boldsymbol{b}_t,
$$

where $\boldsymbol{w}_i$ is the $i^{\text{th}}$ row of $W$ and $\boldsymbol{b}_t^\top$ is the $t^{\text{th}}$ column of $\boldsymbol{B}^\top$. If we let $\tilde{\mu}_{p, \omega} = \mathbb{E}_{\tilde{\boldsymbol{Z}} \sim q_{\psi}}[\tilde{z}_{p, \omega}]$, and correspondingly let $\tilde{\boldsymbol{\mu}}$ be the $P \times \mathcal{F}$ matrix of $\tilde{\mu}_{p, \omega}$'s then $E_{\tilde{\boldsymbol{Z}} \sim q_{\psi}}[x_{i,t}] = \boldsymbol{w}_i^\top \tilde{\boldsymbol{\mu}} \boldsymbol{b}_t$. We may equivalently write $x_{i,t}$ as

$$
\begin{aligned}
x_{i,t} = \text{vec}(x_{i,t}) &= \text{vec}(\boldsymbol{w}_i^\top \tilde{\boldsymbol{Z}} \boldsymbol{b}_t) \\
&= \text{vec}(\boldsymbol{b}_t^\top \tilde{\boldsymbol{Z}}^\top \boldsymbol{w}_i) \\
&= (\boldsymbol{w}_i^\top \otimes \boldsymbol{b}_t^\top) \tilde{\boldsymbol{z}},
\end{aligned}
$$

where $\tilde{\boldsymbol{z}} = \text{vec}(\tilde{\boldsymbol{Z}}^\top)$. We may therefore we may derive the variance of $x_{i,t}$ as

$$
\begin{aligned}
\text{Var}_{\tilde{\boldsymbol{Z}} \sim q_{\psi}}[x_{i,t}] &= \text{Var}_{\tilde{\boldsymbol{Z}} \sim q_{\psi}}[(\boldsymbol{w}_i^\top \otimes \boldsymbol{b}_t^\top) \tilde{\boldsymbol{z}}] \\
&= \mathbb{E}_{\tilde{\boldsymbol{Z}} \sim q_{\psi}}[((\boldsymbol{w}_i^\top \otimes \boldsymbol{b}_t^\top) \tilde{\boldsymbol{z}})^2] - \mathbb{E}_{\tilde{\boldsymbol{Z}} \sim q_{\psi}}[(\boldsymbol{w}_i^\top \otimes \boldsymbol{b}_t^\top) \tilde{\boldsymbol{z}}]^2 \\
&= \mathbb{E}_{\tilde{\boldsymbol{Z}} \sim q_{\psi}}[\text{Trace}[(\boldsymbol{w}_i \otimes \boldsymbol{b}_t)(\boldsymbol{w}_i^\top \otimes \boldsymbol{b}_t^\top) \tilde{\boldsymbol{z}} \tilde{\boldsymbol{z}}^\top]] - (\boldsymbol{w}_i^\top \tilde{\boldsymbol{\mu}} \boldsymbol{b}_t)^2 \\
&= \text{Trace}[(\boldsymbol{w}_i \otimes \boldsymbol{b}_t)(\boldsymbol{w}_i^\top \otimes \boldsymbol{b}_t^\top)(\boldsymbol{V} + \text{vec}(\tilde{\boldsymbol{\mu}})\text{vec}(\tilde{\boldsymbol{\mu}})^\top] - (\boldsymbol{w}_i^\top \tilde{\boldsymbol{\mu}} \boldsymbol{b}_t)^2 \\
&= (\boldsymbol{w}_i^\top \otimes \boldsymbol{b}_t^\top) \boldsymbol{V} (\boldsymbol{w}_i \otimes \boldsymbol{b}_t),
\end{aligned}
$$

where $\boldsymbol{V}$ is the diagonal posterior covariance of $\tilde{\boldsymbol{z}}$ whose elements are the encoded Fourier variational variances, $\tilde{\sigma}_\omega^2(\boldsymbol{Y})$. Therefore, we observe that under the variational posterior $x_{i,t}|\boldsymbol{Y} \sim \mathcal{N}(m_{i,t}, v_{i,t})$, where $m_{i,t} \equiv \boldsymbol{w}_i^\top \tilde{\boldsymbol{\mu}} \boldsymbol{b}_t$ and $v_{i,t} \equiv (\boldsymbol{w}_i^\top \otimes \boldsymbol{b}_t^\top) \boldsymbol{V} (\boldsymbol{w}_i \otimes \boldsymbol{b}_t)$.

We note that for $\lambda_{i,t} = e^{x_{i,t}}$ follows a long-normal distribution, meaning that, for $x_{i,t} \sim \mathcal{N}(m_{i,t}, v_{i,t})$ then $\mathbb{E}[\lambda_{i,t}] = e^{m_{i,t}+\frac{1}{2}v_{i,t}}$. This allows us to specify the posterior expectation of the Poisson likelihood in closed form. Namely,

$$\mathbb{E}_{\tilde{\boldsymbol{Z}}\sim q_\psi}\left[\log \mathcal{P}(y_{i,t}|f(x_{i,t}))\right] = \mathbb{E}_{\tilde{\boldsymbol{Z}}\sim q_\psi}\left[y_{i,t}\log\lambda_{i,t} + \lambda_{i,t}\right] - \log y_{i,t}!$$
$$= y_{i,t}\mathbb{E}_{\tilde{\boldsymbol{Z}}\sim q_\psi}[x_{i,t}] + \mathbb{E}_{\tilde{\boldsymbol{Z}}\sim q_\psi}[\lambda_{i,t}] + \text{const}_{\tilde{\boldsymbol{Z}}}$$
$$= y_{i,t}m_{i,t} + e^{m_{i,t}+\frac{1}{2}v_{i,t}} + \text{const}_{\tilde{\boldsymbol{Z}}}$$

