# OpenReview forum: "Multi-modal Gaussian Process Variational Autoencoders for Neural and Behavioral Data"
_ICLR.cc/2024/Conference — ICLR 2024 poster_

### Official Review · Reviewer_9t6f · 2023-10-30

**Soundness:** 4 excellent
**Presentation:** 4 excellent
**Contribution:** 4 excellent
**Rating:** 8
**Confidence:** 4

**Summary:**

This paper proposes a model for multi-modal data based on a latent space using both shared and mode-specific latent subspaces and Gaussian Process priors. These GP priors are intended to capture smooth variations in the latent space, akin to Gaussian Process Factor Analysis (GPFA) for neural time series data. Another key ingredient is the use of a Fourier basis in the latent space, which allows for a natural low-frequency regularization on latent dynamics. Experiments include both simple synthetic data setups and applications to two real data sets in which neurophysiological recordings are combined with behavior.

**Strengths:**

- Use of GP priors to smooth latent trajectories is a well-matched strategy for smoothing highly noisy neural data.
- Moving to the Fourier basis is a clever idea that allows for a very natural regularization.
- The idea of shared and specific latent subspaces has been done before but adds to the interpretability of the model.
- Several tricks are combined here that render the notoriously inefficient GP inference to be performed efficiently.
- Good set of experiments on real neural data.

**Weaknesses:**

- The synthetic experiments tend to focus on smooth, continuous variation in "incidental" properties (e.g., size and angle of a digit) that may not be a good approximation in many real data sets.
- The GP model may not work well in cases where data do not have a natural trial structure or produce very long behavioral bouts.

**Questions:**

- As noted above, the GP method would seem to need data to be broken into snippets of reasonable size. For distinct trials, this is pretty obvious, but what if the time series are very long (e.g., natural behavior)? Does it work to simply snip the behavior at random? Does the model handle this? Do all snippets need to be the same size?
- Is it possible to put some shrinkage priors on the $W$ matrices in (4) so that unneeded dimensions are pruned away? Can the model distinguish between "true" and "false" shared latents? Is there some notion of parsimony here?

---

> ### Author Response · Authors · 2023-11-19
>
> We thank the reviewer for their positive appraisal of our work. Below we address each of the questions and points raised by the reviewer.
>
> ### Weaknesses:
> > The synthetic experiments tend to focus on smooth, continuous variation [...]
>
> One of the main assumptions that we make with our model is that we assume these multimodal measurements have an underlying low-d smooth structure. Therefore, we agree with the reviewer that MM-GPVAE might not be suitable for multi-modal data that does not smoothly evolve in time (indeed, this is a limitation of GPVAE and GPFA models as well). We talk about limitations of our prior assumptions in the conclusion section of our manuscript to compare other approaches and limitations of other prior approaches. However, comparing multi-modal neuroscience models with different prior assumptions remains an important avenue for exploration.
>
> ### Questions:
> > As noted above, the GP method would seem to need data to be broken into snippets of reasonable size. [...]
>
> That is correct. For our evaluations on real datasets, we need to artificially split our data into trials of equal length, which we describe in more detail in the appendix, page 8 and 9.
>
> We find that using the Fourier parameterization aids with this process, and the model is able to handle longer-time series trials if processed in the fourier domain (likely due to the lack of the need to invert the prior GP covariance matrix, and the overall reduction in the number of variational parameters)
> > Is it possible to put some shrinkage priors on the  matrices in (4) [...]
>
> The introduction of shrinkage priors is a good idea, and an avenue we are pursuing for follow-up work. For now, we avoid unneeded dimensions through our process of cross-validation. As we indicated in our other responses, we have added an additional section in the appendix devoted to explaining this process (see "Latent dimensionality selection" on page 9 in appendix). Briefly, we assure we have no more dimensions than needed by first assessing the dimensionality per modality, and then increasing the number of shared dimensions one by one while evaluating cross-validated performance. In all tested datasets, using at least 1 shared dimension performed better than 0 shared dimensions, suggesting some cross-modality structure. We increased the number of shared dimensions sequentially until we no longer saw an increase in cross-validated performance, and used that to identify the latent dimensionality. This process, however, is somewhat intensive and a shrinkage prior would be very helpful in speeding up this procedure.
>
> We again thank the reviewer for their thoughtful comments and suggestions, and for their time reviewing the paper. We hope that the revisions and responses in our updated paper have sufficiently addressed all of the concerns and questions raised by the reviewer. If there are still any questions or concerns, we would be happy to clarify them and engage in further discussion.

---

> > ### Comment · Reviewer_9t6f · 2023-11-20
> >
> > I appreciate the authors' thorough responses to all reviewers. I am convinced this is a valuable contribution to both the neuroscientific data analysis and multimodal VAE literatures and will be maintaining my score.

---

### Official Review · Reviewer_y53H · 2023-10-31

**Soundness:** 2 fair
**Presentation:** 1 poor
**Contribution:** 2 fair
**Rating:** 3
**Confidence:** 4

**Summary:**

The paper proposes a variational auto encoder framework to jointly model neural activities and behavior data. Their contribution is to add Gaussian Process Prior and parameterize the latent in Fourier domain.  Their goal is to distinguish the separate or joint latents from multiple modalities, and assume a linear combination of those latents. The work is evaluated on one simulated dataset (MNIST + spike train) and two real-world datasets from different animal species (drosophila, Manduca).

**Strengths:**

1. The work is well-motivated,  jointly train latent variable models from multi-modal neural data is an important question.
2. The contribution to use Gaussian Process prior to capture correlations in latent space, and parameterize the latent in Fourier space are novel contributions.
3. The work is demonstrated on multiple datasets to improve its soundness, including one synthetic dataset and two real-world datasets from different animal species .

**Weaknesses:**

Methods:
1. The assumption of linearity between latents in eqn (4) seems over-simplified, other nonlinearity variants could be further investigated.
2. The construction of simulated dataset is confusing, it is not clear to see how the rotation angle of MNIST image is inherently related to spiking distribution. And it is puzzling that how it is tightly linked with real-world datasets.
3. The paper proposed parameterize the latents in Fourier space to constrain them more distinguishable, and attempt to prune the high-frequency components. While it didn't provide solid ablation studies and compared to the original temporal space on real datasets, and it is problematic to claim no information preserved in high-frequency domains.
4. The authors didn't provide clear explanation or guidance about important hyperparameters of latent dimensions.

Evaluations:
1. No quantitative measurements on real datasets only with a few qualitative samples, and not compared with other baselines.
2. In Fig 4(d), different behavior states are not clearly separable in these subspaces.
3. Blurriness and mismatch of reconstruction outputs and latents in the results.

Writing:
1. The paper is not well organized, missing quantitative evaluations and comparisons with baselines, a few important contents are pointed to supplementary, while it is not clearly described there either.

**Questions:**

1. Is nonlinearity needed in eqn (4)?
2. How to decide dimensions for separate and joint latent?

---

> ### Author Response · Authors · 2023-11-19
>
> ### Methods:
>
> 1. Equation 4 simply functions to combine the shared and modality specific latents in the first step of the decoder. This is akin to the latent space in probabilistic CCA models (see for example equation 2 in Gunderson et al. 2019) and so it acts to partition the variability between the shared and independent latent subspaces. This linear operation is therefore an essential feature of our multi-modal model.
>
> - Gundersen, Gregory, et al. "End-to-end training of deep probabilistic CCA on paired biomedical observations."
> 2.
>
> > The construction of simulated dataset is confusing,
>
> We apologize for this confusion. Please see our response to reviewer YQRh, where we clarify this in detail. If you have any additional questions on this, please let us know.
>
> > [...]  linked with real-world datasets.
>
> We draw the reviewers attention to the final figure, where it is precisely the case that the latent variables that describe the neural activity are a linear combination of latents that are shared with a high dimensional moving stimulus (closely related in structure to the MNIST image) and independent wing-flapping (like our other 1d GP neural-only latent in our simulated data). By validating our approach in this synthetic setting first, we can be more confident in understanding these uncovered latents in figure 5.
> To re-emphasize this real-world finding, the neural activity is tuned from a two dimensional latent space, one dimension of which tracks the movement of the 2d visual stimulus, and the other dimension corresponds to wing-flapping. The overall Poisson rate for each neuron is thus a linear combination of these two experimental features, in a similar form to how we construct our synthetic data.
>
> 3. Thank you for this suggestion. We have included the Fourier approach without pruning the frequencies in the appendix (see figure 6). As you can see, not pruning the frequencies prevents us from identifying the correct smoothly evolving latent structure, and so we did not continue with an unpruned approach in this work.
>
> Importantly, we do not claim that there is no high-frequency information in the latent space (there very likely is) but our pruning regularizes our model in such a way to be able to uncover meaningful latents with clear temporal structure. As the reviewer suggests, our model is thus limited in its ability to identify fast moving temporal structure, and we will be sure to emphasize this limitation more clearly in the final manuscript.
>
> 4. We apologize for this not being more clear. We mentioned the latent dimension selection process in the appendix but agree our treatment was cursory. We clarify our approach to reviewer Kdv4, so please read this response for more information. We have additionally clearly described our dimensionality-selection method in the appendix, now it can be found under ‘latent dimensionality selection’ section.
>
> Regarding other model hyperparameters, we have a section devoted to the observation variance parameter in the supplement, as well as a thorough description of our selection of length scales for the GP latents and neural network architectures. In general, model performance is not sensitive to specific network architectures and length scale initializations can encourage smoother or less smooth after inference is complete to some extent (also see page 4, 6, 7, 9, and 10 of the appendix for more information).
> Please let us know if you need any further clarification on any specific model hyperparameters.

---

> > ### Comment · Reviewer_y53H · 2023-11-23
> >
> > Thanks to the authors for adding experiment results and clarifications. I think part of my concerns are addressed. While I think this synthetic dataset is still oversimplified compared to the real-world datasets, it fails to demonstrate the real capabilities of the proposed methods. The added ablation studies and baselines are still performed in simulated data, not real data. Comprehensive evaluations require more datasets and baselines (i.e. existing benchmarks). As mentioned by other reviewers, the novelty of this work is also limited given it is a combination of existing methods (i.e. Fourier representation).

---

> > > ### Author Response · Authors · 2023-11-23
> > >
> > > > [...] synthetic dataset is still oversimplified [...]
> > >
> > > We would like to draw the reviewers attention to our previous response (Methods, Section 2, starting with 'We draw the reviewers attention to the final figure ...') where we talked about how the synthetic dataset can be related to real world dataset.
> > >
> > > To add to this; We believe that our synthetic dataset concisely demonstrates what MM-GPVAE achieves, where we can relate the underlying latents to our modalities. We find that our dataset construction aligns with other multi-modal models in neuroscience. If the reviewer would like to point out a specific paper or dataset that they think is more suitable, we would be happy to take a look. We also invite the reviewer to take a look at the construction of synthetic datasets in the following related papers which will show that our synthetic dataset construction is nearly ubiquitous for this class of models and not oversimplified:
> > >
> > > Yu, B., Cunningham, J., et al. (2009). Gaussian process factor analysis for low-d single-trial analysis of neural population activity.
> > >
> > > Duncker, L. and Sahani, M. (2018). Temporal alignment and latent gaussian process factor inference in population spike trains.
> > >
> > > Zhao, Yuan, and Il Memming Park. Variational latent gaussian process for recovering single-trial dynamics from population spike trains.
> > >
> > > Keeley, Stephen, et al. Identifying signal and noise structure in neural population activity with Gaussian process factor models.
> > >
> > > Jensen, Kristopher, et al. Scalable Bayesian GPFA with automatic relevance determination and discrete noise models.
> > >
> > > Casale, F. P., et al. (2018). Gaussian process prior variational autoencoders. Advances in neural information processing systems, 31.
> > >
> > > Ramchandran, S., et al. (2021). Longitudinal variational autoencoder. International Conference on Artificial Intelligence and Statistics (pp. 3898-3906). PMLR.
> > >
> > > PSID - Sani et al. (2021) "Modeling behaviorally relevant neural dynamics enabled by preferential subspace identification"
> > >
> > > TNDM - Hurwitz et al. (2021) “Targeted Neural Dynamical Modeling”
> > > >  The added ablation studies and baselines are still performed in simulated data, not real data.
> > >
> > > As we mention in the manuscript in our evaluation of the hawkmoth dataset, the shared latent accounts for about 30% of the variability of the image reconstructions, and about 5% of the variability of the neural data. As we point out, this is expected as the neural data is dominated by wing-flapping. We have since looked at what features of the stimulus and neural activity are encoded in the shared subspace by removing this shared component and plotting the data reconstructions. We find that removing the shared component removes the slowly varying features of the neural reconstructions and blurs and distorts the overall image reconstruction, particularly at the times when the stimulus changes direction. If the reviewer feels strongly, we could add these figures to the appendix.
> > > > [...] the novelty of this work is also limited given it is a combination of existing methods (i.e. Fourier representation).
> > >
> > > We provided a thorough answer to this in our response to reviewer Kdv4. For the reviewers convenience we provide the same answer below:
> > >
> > > We believe that we are the first time-series latent variable model developed that uses this partitioning to separate both shared and independent latent trajectories in multi-modal settings in neuroscience, and we do so in a way that is more flexible and scalable compared to competing multi-modal methods (see appendix). We also would like to highlight our novel amortization method in the Fourier domain for a deep GP model, as well as our derivation of a model specific ELBO in the appendix that precludes the need to approximate two of the terms in the objective.
> > >
> > > Though a Fourier representation of the GP is used and is standard in the GP literature, as far as we know they have never been implemented in a deep setting, which requires not just a new parameterization of the nonlinear model but a distinct inference approach. This new inference approach (the amortization of Fourier frequencies) in our model differs from other GP models as well as other VAE models. To be clear, the contribution is that we encode a time-domain representation, and then transform it into the Fourier space prior to decoding back into the time domain. This extra encoding and decoding stages markedly scales and stabilizes inference despite being an additional linear step as it provides the GPVAE with a strong regularization.
> > > This also allows us to modify the existing GPVAE to accurately identify the latent space, as evidenced by our simulated examples. This type of pruned variational posterior and amortized encoding implementation would not be readily available elsewhere or in any standard GP packages.
> > >
> > > If the reviewer could point out which combination they are referring to and how those papers include our contributions, we would be happy to discuss this further.

---

> ### Author Response · Authors · 2023-11-19
> **Cont. for Evaluations, Writing, and Questions**
>
> ### Evaluations:
>
> 1.
>
> > No quantitative measurements on real datasets
>
> Please see our response to reviewer bxWt, as we emphasize our quantitative measurements about the contributions of the shared and independent subspaces on the hawkmoth dataset, and propose the addition of additional reconstruction examples of data from each modality as we ablate the shared and independent latents in the appendix.
>
> > Not compared with other baselines
>
> As we mention to other reviewers, we do compare to two time-series multi-modal models used in neuroscience today in the appendix, both on a synthetic dataset as well as a real neural dataset (fig 1 and 2). However, it is clear we did not highlight this enough in the main body of the manuscript, and so we will be sure to update this accordingly.
>
> 2. That is correct, the MMGPVAE does not separate behavioral conditions particularly cleanly across fly behaviors. The separation of behavioral categories of the latents is an avenue for further scientific exploration, and may require tuning additional model features or adjusting the geometry of the latent prior. However, figure 4 is meant to demonstrate that the MMGPVAE is able to generate good predictive performance on this multi-modal dataset in held-out trials, and that each of the latent subspaces are able to be visualized and analyzed in this setting. (as far as we know, we are the first multi-modal model flexible enough to describe simultaneous 16-dimensional limb-tracking and calcium recordings.)
>
> 3. The blurriness of the 3 in our model reconstructions is expected given our model. To be clear, our goal is not to best reconstruct the three, but rather to identify the smoothly evolving angle and scale of the three in the latent space. The reconstructed threes are thus provided only angle and scale information from the latents, and so they otherwise reflect an average of all handwritten threes in the training set (hence the blurry, canonical 3 looking the same at all angles and scales). This is an essential point in our work, and we are happy to clarify further if the reviewer has additional questions or concerns. (note also that the blurry canonical 3 is seen in other GPVAEs of Casale et al, Fortuin et al. and Ramchandran et al).
> ### Writing:
>
> We apologize for any confusion in our work. If the reviewer can be more specific about which important contents they feel are insufficiently addressed beyond what is discussed above, please let us know.
> ### Questions:
> 1. A nonlinear operation is used as the next step in each of our decoders (an exponential nonlinearity to generate Poisson rates for the GPFA description of neural data and a deep neural network to describe the behavior/stimulus), so the ultimate decoding mapping to data is a nonlinear operation. If this first step were nonlinear, it would be highly unusual, a more difficult to train and we would lose the ability to interpret our shared/independent latent partitioning. (see also Gunderson et al 2019 for additional context).
>
> 2. We thank the reviewer (and other reviewers) for pointing this out and have more clearly highlighted this procedure in the Appendix, page 9. Please let us know if further clarification is needed.
>
>
> We hope that the revisions and responses in our updated paper have sufficiently addressed all of the concerns and questions raised by the reviewer. If this is the case, we hope the reviewer could consider updating their score accordingly. If there are still any questions or concerns, we would be happy to clarify them and engage in further discussion.

---

### Official Review · Reviewer_bxWt · 2023-11-05

**Soundness:** 2 fair
**Presentation:** 4 excellent
**Contribution:** 3 good
**Rating:** 8
**Confidence:** 4

**Summary:**

This paper introduces a method, MM-GP-VAE, for modeling multimodal data, presented in the context of analyzing neural and behavioral data.  The balance that the proposed method strikes is retaining the interpretability and disentanglement enjoyed by simpler models, but retaining the expressivity and ease-of-use of deep methods.  Low-dimensional latent states describe each channel individually, and a shared low-dimensional state describes the shared covariance.  The recurrent model is implemented as a Fourier-domain GP-VAE, to encourage smoothness in the latent state and increase expressivity.  The model is benchmarked on a synthetic MNIST-like experiment, and is then applied to neural data.  The model appears to perform well against some sensible baselines.

**Strengths:**

The paper is incredibly well presented.  Figures are prepared exceptionally well.  The prose is clear, and presents a self-contained introduction to all of the necessary techniques and considerations.

The method itself is gracefully simple.  Although there is not a huge methodological contribution, the correct components parts are assembled and deployed in a way that is very experimentally useful.  I could see this methodology having real uptake within the community, and engendering multiple follow-up works.

**Weaknesses:**

I do have several queries/concerns however:

- **a. Fixed time horizon**:  The use of an MLP to convert the per-timestep embeddings into per-sequence Fourier coefficients means that you can only consider fixed-length sequences.  This seems to me to be a real limitation, since often neural/behavioral data – especially naturalistic behavior – is not of a fixed length.  This could be remedied by using an RNN or neural process in place of the MLP, so this is not catastrophic as far as I can tell.  However, I at least expect to see this noted as a limitation of the method, and, preferably, substitute in an RNN or neural process for the MLP in one of the examples, just to concretely demonstrate that this is not a fundamental limitation.

- **b. Hidden hyperparameters and scaling issues**:  Is there a problem if the losses/likelihoods from the channels are “unbalanced”?  E.g. if the behavioral data is 1080p video footage, and you have say 5 EEG channels, then a model with limited capacity may just ignore the EEG data.  This is not mentioned anywhere.  I think this can be hacked by including a $\lambda$ multiplier on the first term of (6) or raising one of the loss terms to some power (under some sensible regularization), trading off the losses incurred by each channel and making sure the model pays attention to all the data.  I am not 100% sure about this though.  Please can the authors comment.

- **c.  Missing experiments**:  There are a couple of experiments/baselines that I think should be added.
  - Firstly, in Figure 3, I'd like to see a model that uses the data independently to estimate the latent states and reconstruction.  It seems unfair to compare multimodal methods to methods that use just one channel.  I’m not 100% sure what this would look like, but an acceptable baseline would be averaging the predictions of image-only and neuron-only models (co-trained with this loss).  At least then all models have access to the same data, and it is your novel structure that is increasing the performance.
  - Secondly, I would like to see an experiment sweeping over the number of observed neurons in the MNIST experiment.  If you have just one neuron, then performance of MM-GP-VAE should be basically equivalent to GP-VAE.  If you have 1,000,000 neurons, then you should have near-perfect latent imputations (for a sufficiently large model), which can be attributed solely to the neural module.  This should be a relatively easy experiment to add and is a good sanity check.
  - Finally, and similarly to above, i’d like to see an experiment where the image is occluded (half of the image is randomly blacked out).  This (a) simulates the irregularity that is often present in neural/behavioral data (e.g. keypoint detection failed for some mice in some frames), and (b) would allow us to inspect the long-range “inference” capacity of the model, as opposed to a nearly-supervised reconstruction task.

  Again, these should be reasonably easy experiments to run.  I’d expect to see all of these experiments included in a final version (unless the authors can convince me otherwise).

- **d.  Slightly lacking analysis**:  This is not a deal-breaker for me, but the analysis of the inferred latents is somewhat lacking.  I’d like to see some more incisive analysis of what the individual and shared features pull out of the data – are there shared latent states that indicate “speed”, or is this confined to the individual behavioral latent?  Could we decode a stimulus type from the continuous latent states?  How does decoding accuracy from each of the three different $z$ terms differ? etc.  I think this sort of analysis is the point of training and deploying models like this, and so I was disappointed to not see any attempt at such an analysis.  This would just help drive home the benefits of the method.


### Minor weaknesses / typographical errors:

1.  Page 3:  why are $\mu_{\psi}$ and $\sigma_{\psi}^2$ indexed by $\psi$?  These are variational posteriors and are a function of the data; whereas $\psi$ are static model parameters.

2.  Use \citet{} for textual citations (e.g. “GP-VAE, see (Casale et al., 2018).” ->   “GP-VAE, see Casale et al. (2018).”)

3.  The discussion of existing work is incredibly limited (basically two citations).  There is a plethora of work out there tackling computational ethology/neural data analysis/interpretable methods.  This notable weakens the paper in my opinion, because it paints a bit of an incomplete picture of the field, and actually obfuscates why this method is so appealing!  I expect to see a much more thorough literature review in any final version.

4.  Text in Figure 5 is illegible.

5.  Only proper nouns should be capitalized (c.f. Pg 2 “Gaussian Process” -> “Gaussian process”), and all proper nouns should be capitalized (c.f. Pg 7 “figure 4(c)”).

6.  Figure 1(a):  Is there are sampling step to obtain $\tilde{\mu}$ and $\tilde{\sigma}^2$?  This sample step should be added, because right now it looks like a deterministic map.

7.  I think “truncate” is more standard than “prune” for omitting higher-frequency Fourier terms.

8.  I find the use of “A” and “B” very confusing – the fact that A is Behaviour, and B is Neural?  I’m not sure what better terms are.  I would suggest B for Behavioural – and then maybe A for neural?  Or A for (what is currently referred to as) behavioral, but be consistent (sometimes you call it “other”) and refer to it as Auxiliary or Alternative data, and then B is “Brain” data or something.

9.  The weakest section in terms of writing is Section 3.  The prose in there could do with some tightening.  (It’s not terrible, but it’s not as polished as the rest of the text).

10.  Use backticks for quotes (e.g. ‘behavioral modality’ -> ``behavioral modality’’).

**Questions:**

I don’t have any further questions from those outlined in Weaknesses.

**Note:** If the authors can allay my concerns, then I fully intend to increase my review score.

---

> ### Author Response · Authors · 2023-11-18
>
> We thank the reviewer for highlighting the relevance of this work to the neuroscience community and praising the approach and experiments. Below we address each of the questions and points raised by the reviewer.
> ### Weaknesses
> > a.
>
> We agree with the reviewer that this is a limitation of the model, and we will be sure to highlight this in the main body of the final manuscript. We do, in fact, need to split our datasets into ‘trials’ of fixed length to fit the model (as is briefly mentioned in the appendix sections on pages 8 and 9). Fortunately, this did not practically prevent our model from providing insight in the datasets in 4 and 5.
> > b.
>
> This is a good point, and is discussed in some detail in the appendix (see page 10 in appendix). To highlight the important point – our learned $\sigma^2$ parameter did not vary much from its initialized value, and so our choice of the initialization of $\sigma^2$ reflects exactly the kind of trade-off the reviewer describes. We therefore initialized the $\sigma^2$ parameter to a value proportional dimensionality of the observed behavioral modality, and this assured a balancing across modalities for any of the real-world datasets we tested.
> > c.
>
> > Firstly [...]
>
> We apologize but we are a little unclear as to this point. Can the reviewer clarify further what they mean here? Our ablations in figure 3 do use the datasets independently, and the latent dimensionality is kept the same per-modality (two dimensions for each unimodal evaluation). In other words, each unimodal piece of our model has identical flexibility whether or not the model is trained jointly or separately. Therefore, it is the cross-modality structure that increases the performance (the neural latents are able to use the structure of the image data to better reconstruct rates, and vice-versa).
> > Secondly [...]
>
> Thank you for this suggestion. We have added a figure, fig 7,  to the appendix showing exactly this. We see that increasing the number of neurons only improves the identification of shared and neural latents, and, expectedly, does not affect the ability of the model to identify the image-only latent structure.  We also see improvement in reconstructions across both modalities as the number of neurons increases
> > Finally [...]
>
> An excellent suggestion. We have added it to the appendix. Our smooth gp-based approach allows for better identification of an uncorrupted image than a standard VAE while retaining the ability to reconstruct neural rates, rendering the MMGPVAE robust to measurement errors.
> > d.
>
> Incidentally, since submitting the manuscript, we have follow-up work that begins to answer this question in the context of the hawkmoth dataset. As we mention in the manuscript in our evaluation of the hawkmoth dataset, the shared latent accounts for about 30% of the variability of image reconstructions, and about 5% of the variability of neural data. As we point out, this is expected as the neural data is dominated by wing-flapping. We have since looked at what features of the stimulus and neural activity are encoded in the shared subspace by removing this shared component and plotting the data reconstructions. We find that removing the shared component removes the slowly varying features of neural reconstructions and blurs and distorts the overall image reconstruction, particularly at the times when the stimulus changes direction. If the reviewer feels strongly, we could add these figures to an appendix, but currently we feel it is more appropriate for follow-up work.
>
> ### Minor weaknesses
> Thank you for the improvements to our text. All the changes will be implemented in the final manuscript. We address some of the more important minor points below:
> > 3.
>
> We will change this section title in the discussion, to “competing multi-modal approaches in neuroscience”, and thank the reviewer for pointing out this confusion. While we survey the field significantly more broadly in the introduction, in this discussion section, we wanted to simply point out two existing multi-modal time-series LVMs that have been used in similar settings in neuroscience. We find it important to point out these models as we compare our approach to them in two figures in the appendix. We will be sure to clarify the writing here and make sure we conclude with a more accurate picture of the field.
> > 8.
>
> We apologize for this confusion. We propose flipping the notational indices, where A reflects “activity” and B reflects “behavior”
>
> We again thank the reviewer for their thoughtful comments and suggestions, and for their time reviewing the paper. We hope that the revisions and responses in our updated paper have sufficiently addressed all of the concerns and questions raised by the reviewer. If this is the case, we hope the reviewer could consider updating their score accordingly. If there are still any questions or concerns, we would be happy to clarify them and engage in further discussion.

---

> ### Comment · Reviewer_bxWt · 2023-11-19
> **Mostly there**
>
> To the authors,
>
> Thank you for your response and clarifications, and for swiftly updating the paper.
>
> In response to "C. Firstly":  As far as I understand, in Figure 3c, the baselines are only provided with one channel, whereas your proposed method is provided with two channels.  Is the performance increase coming from simply getting to see both channels, whereas the baselines only see one channel?  Are the authors saying that "image-only GP-VAE" is in some way accessing the neural data?
>
> E.g., You're making a salad.  It is unfair to compare a salad made by someone who just had access to tomatoes, to someone who just had lettuce, to someone who gets to use both tomatoes and lettuce.  I want you to compare four salads (models): just tomato (neuron only GPFA), just lettuce (image only GP-VAE), lettuce and tomato side by side but no mixing (the baseline I'm requesting, e.g. averaging predictions/reconstructions of neuron only GPFA and image only GP-VAE), and a salad with lettuce and tomato all mixed together (MM-GPVAE).*
>
> Please correct me if my understanding of the experiment presented is incorrect.
>
> In response to "D.":  I do feel reasonably strongly on this, actually.  If you have such results, then I think at least a taster of them should be included here.  If you are angling _this_ paper as an interpretable neuroscience method, then I think some interpretations/neuroscience should be included.  I don't think you should include so much analysis that you cannot publish another paper somewhere else, but I think this paper might've been "sliced a bit thin" in that context.  I would very much like to see some simple analysis in this vein included in the final version.
>
> Overall, I think this paper is of publication quality.  Subject to the inclusion of a more thorough discussion of related techniques, and the comparisons suggested by Kdv4, i vote for the inclusion of this paper.
>
> Thank you very much, and good work,
>
> bxWt
>
> _*This is the most ridiculous thing I've written in a review.  I hope you enjoyed it and/or found it useful...!_

---

> > ### Author Response · Authors · 2023-11-22
> >
> > We thank the reviewer for the prompt reply. We will include an additional figure at the end of the appendix that shows two example neurons in the hawkmoth dataset that show a high degree of tuning to the stimulus tracking for the final version of the paper. Here, we can generate these same neural rates from just the stimulus latents in one case  and just the neural latents in another. We see here a dissociation of the two features of these neurons (wing-beating and stimulus tracking). When the neuron rate contains only the neural-independent component, the fast oscillation corresponding to motor movement is the dominant signal. When the neuron rate contains only the shared component, it exhibits a slow modulation like that of the moving stimulus. Together, these features combine to give the overall rate of these neurons.
> >
> > Regarding the additional evaluation, we will re-run our unimodal benchmarks for the final paper version where we include additional examples where the amortization of the variational distribution includes encoding from both modalities, and the only distinction will be in the reconstruction likelihood terms in the ELBO. That is, we run “GPFA-only” and “GPVAE-only” but in each case the models are free to generate their variational parameters using neural network encoders from data from both modalities. (to borrow the reviewers metaphor, the lettuce-only chef has access to both lettuce and tomatoes).
> >
> > We actually have previously explored this, and we’ve seen no difference in the reconstructions of the data or the latents if we expand the variational encoding modalities. This is because the component of the model that drives inference and latent identification is the likelihood distributions, not the flexibility of the variational representation. This is somewhat expected given the objective form –  I.e. GPFA-only can only reconstruct data as well as its generative model prescribes, irrespective of how flexible the estimated variational posterior parameterization is. Regardless, we will be sure to include the results in our final version of the paper. We again thank the reviewer for their thoughtful comments and suggestions.

---

### Official Review · Reviewer_YQRh · 2023-11-05

**Soundness:** 2 fair
**Presentation:** 2 fair
**Contribution:** 2 fair
**Rating:** 5
**Confidence:** 3

**Summary:**

This paper proposes a multi-modal gaussian process VAE model for neuroscientific data. Due to the time-varying nature of neuronal data, the authors use gaussian priors for Fourier frequencies of the data. Thus the network features FFT and IFFT layers. Results are shown on synthetic data as well as on calcium signaling data from Drosophila and Moth EMG data.

**Strengths:**

The use of Fourier frequencies as latent variables can have some advantages in learning neural features that correlate with motor or other behaviors. The authors also separate shared from independent variability in the neural data which can be useful in understanding what part of behavior could actually be predicted from specific neuronal measurements.

**Weaknesses:**

-The synthetic data seems weak and unnecessary. Why not use a simulator like NEST or augment one and create "behavioral" data with known connections to this. MNIST images have a different structure because they are a two dimensional image.
-The authors lack comparison to anything but an ablation of their own model, and that too on the above artificial dataset.
-At least the authors should compare to RNN/transformers/ODE or other sequential models for this data
-The authors seem to criticize DNNs for transforming the data too much such that the latent variables are "not obviously related" to external variables, and yet they too use DNNs.
-There is no theory or study of identifiability of the dynamics or conditions/noise under which dynamics are recovered.

**Questions:**

The synthetic data does not make sense to me, what exactly is the connection between the digits and the "spikes"?

---

> ### Author Response · Authors · 2023-11-19
>
> We thank the reviewer for finding MM-GPVAE useful for understanding the components of behavioral and neural data. Below we address each of the questions and points raised by the reviewer.
>
> ### Weaknesses:
> > [...] simulator like NEST [...]
>
> Assessing performance for GPFA models in neuroscience to synthetic poisson or gaussian spiking data generated from a low-dimensional subspace is nearly ubiquitous for this class of models (e.g. see Yu et al 2009, Lakshmanan et al 2015, Zhao and Park 2017, Dunker and Sahani 2018, Keeley et al 2020, Jensen et al 2021, Gokcen et al 2022) A simulator like NEST is not designed to generate spikes with true low-dimensional generative latent structure so unless we’ve misunderstood the reviewer we would have no way of validating our LVM approach on such synthetic data. Similarly, GPVAE models each use rotating digits as benchmarks for their model performance (see Casale et al 2018, Fortuin et al 2020, Ramchandran et al 2021) generated from a low-dimensional latent space. Because, in each of these cases in these works, true generative low-dimensional GP latents give rise to very different kinds of data (MNIST and Poisson spikes), we consider this precise type of synthetic data vital in validating our inference approach, and necessary to relate our model pieces to the GPFA and GPVAE literature upon which we draw.  By using rotating digits and GPFA-generated spike trains, we are able to demonstrate that we maintain state-of-the-art performance on the relevant datasets for each unimodal model (GPFA and GPVAE). This would be more difficult to do if we generated a completely new type of synthetic dataset.
> > MNIST images have a different structure[...]
>
> MNIST images have a different structure than our experimental validation in figure 4 but quite similar structure to our experimental validation in figure 5, where we assess the model on spiking motor neurons and a smoothly moving 2d visual stimulus.
> > The authors lack comparison [...]  authors should compare to RNN/transformers/ODE [...]
>
> We apologize if we did not highlight these clearly enough in the main manuscript, but we do, in fact, compare MM-GPVAE to an RNN based model (Hurwitz et al.) as well as an ODE based model (Sani et al.). Both of these existing models are multi-modal models used in neuroscience . We evaluate these existing multi-modal models on both synthetic and real-world datasets in the figures 1 and 2 of the appendix.
>
> - Sani et al. (2021) "Modeling behaviorally relevant neural dynamics enabled by preferential subspace identification"
>
> - Hurwitz et al. (2021) “Targeted Neural Dynamical Modeling”
>
> > The authors seem to criticize DNNs [...]
>
> We are of course not critical of DNNs in general. As the reviewer points out, there are many DNNs used in our work, including in the encoding networks in all models, and in the decoding networks for behavioral data modalities. These networks are crucial for model inference and greatly enhance the flexibility of our approach. However, DNNs are limited in their abilities to provide scientific insight, and this is precisely what motivates the use of a linear decoder for the neural data. By having a linear decoder, our map from latents to rates is akin to GPFA, allowing the MMGPVAE to capitalize on its wide successes (see the GPFA citations below). In addition, it  allows us to perform the analyses in figure 5 of the manuscript, where we can evaluate how individual neurons are independently tuned to shared (stimulus-tracking) and independent (wing-flapping) subspaces. We are pursuing follow-up work which uses these linearly-identified tuning features to better understand the hawkmoth motor system.
>  > There is no theory [...]
>
> This is correct. We will be sure to highlight this limitation in the final version of the manuscript, and we thank the reviewer for pointing this out.

---

> > ### Author Response · Authors · 2023-11-19
> > **Cont. for Questions and Citations**
> >
> > ### Questions:
> > > [...] connection between the digits and the "spikes"?
> >
> > The smoothly varying angle of the digit images is generated from a single dimensional latent variable drawn from a Gaussian process (the same latent form that is seen in the original Casale et al 2018, but confined here to be one-dimensional). This same latent variable also represents one latent dimension of neural population activity (the same form that is seen in Zhao and Park 2017). By sharing this latent dimension, we are generating synthetic data where some of the neural spiking is tuned to the stimulus angle, and an additional 1d GP reflects the portion of the neural variability that is not tuned to the rotating image. Our goal is, from the spiking and image data alone, to uncover the component of neural activity that is related to (or tuned to) the rotating image. That is, we seek to find the 1d latent subspace that explains the variability of both datasets across time. We find that we are able to uncover the 1d latent in the neural activity that is tuned to the angle of the rotating image. (this is similar, for example, to our real-world example in figure 5, where 1d latent of neural activity is tuned to a visual stimulus position). If this is not clear, we are happy to answer any follow-up questions.
> >
> > We hope that the revisions and responses in our updated paper have sufficiently addressed all of the concerns and questions raised by the reviewer. If this is the case, we hope the reviewer could consider updating their score accordingly. If there are still any questions or concerns, we would be happy to clarify them and engage in further discussion.
> >
> >
> > ### GPFA models:
> > - Yu, B., Cunningham, J., Santhanam, G., Ryu, S., Shenoy, K., and Sahani, M. (2009). Gaussian process factor analysis for low-d single-trial analysis of neural population activity.
> > - Duncker, L. and Sahani, M. (2018). Temporal alignment and latent gaussian process factor inference in population spike trains.
> > - Lakshmanan, Karthik C., et al. "Extracting low-dimensional latent structure from time series in the presence of delays."
> > - Zhao, Yuan, and Il Memming Park. "Variational latent gaussian process for recovering single-trial dynamics from population spike trains."
> > - Keeley, Stephen, et al. "Identifying signal and noise structure in neural population activity with Gaussian process factor models."
> > - Jensen, Kristopher, et al. "Scalable Bayesian GPFA with automatic relevance determination and discrete noise models."
> > - Gokcen, Evren, et al. "Disentangling the flow of signals between populations of neurons."
> >
> > ### GPVAE models:
> > - Casale, F. P., Dalca, A., Saglietti, L., Listgarten, J., & Fusi, N. (2018). Gaussian process prior variational autoencoders. Advances in neural information processing systems, 31.
> > - Fortuin, V., Baranchuk, D., Rätsch, G., & Mandt, S. (2020). Gp-vae: Deep probabilistic time series imputation. In International conference on artificial intelligence and statistics(pp. 1651-1661). PMLR.
> > - Ramchandran, S., Tikhonov, G., Kujanpää, K., Koskinen, M., & Lähdesmäki, H. (2021). Longitudinal variational autoencoder. In International Conference on Artificial Intelligence and Statistics (pp. 3898-3906). PMLR.

---

### Official Review · Reviewer_Kdv4 · 2023-11-08

**Soundness:** 3 good
**Presentation:** 3 good
**Contribution:** 2 fair
**Rating:** 5
**Confidence:** 4

**Summary:**

The authors proposed a multi-modal Fourier-domain Gaussian Process variational autoencoder (MM-GPVAE), aiming at jointly modeling observed neural and behavioral data in neuroscience applications with both shared and independent latent subspaces. MM-GPVAE combines the ideas in previous GP-VAE, GP Factor Analysis (GPFA), and spectral representation. The authors have also presented simulations and two neuroscience case studies showing the effectiveness of MM-GPVAE modeling neural measurement / spiking data with either visual stimulus or position data.

**Strengths:**

1. This submission exploits the multi-modal latent variable models based on GP-VAE, using Poisson and normal likelihoods for spiking and image data respectively, which can help modeling both shared and independent factors across modalities in neuroscience.

2. Experiments on simulated and real-world datasets with visualized results showing the potential in jointly analyzing neuroscience data.

3. The presentation is clear with articulated motivations of the presented mm-GPVAE. Source code is also provided.

**Weaknesses:**

1. Methodological contributions may be limited. Many papers have done similar things, such as extending the latent space into frequency domain [1,3] and using the Poisson model for spiking data [1,2]. The method proposed in this paper is combination of existing methods, not fundamentally different.

2. The experimental results can be more comprehensive: (1) More baselines, especially similar methods such as [1,4] need to be compared; (2) The authors stated one main advantage of the Fourier domain is that high frequencies can be pruned and thus we can sparsify the variational parameters. However, no results are provided about this point; (3) An ablation study between the models with Frequency prior and with other priors (e.g., a temporal form) should be added; (4) More discussions on the claimed interpretability should be provided.

[1] Keeley S, Aoi M, Yu Y, et al. Identifying signal and noise structure in neural population activity with Gaussian process factor models. Advances in neural information processing systems, 2020, 33: 13795-13805.

[2] Duncker L, Sahani M. Temporal alignment and latent Gaussian process factor inference in population spike trains. Advances in neural information processing systems, 2018, 31.

[3] Hensman J, Durrande N, Solin A. Variational Fourier Features for Gaussian Processes. J. Mach. Learn. Res., 2017, 18(1): 5537-5588.

[4] Pearce M. The Gaussian Process prior VAE for interpretable latent dynamics from pixels. Symposium on advances in approximate Bayesian inference. PMLR, 2020: 1-12.

**Questions:**

1. How sensitive is the performance with respect to the setting of several hyperparameters, including $F$ and the dimension of shared and independent latent subspace, etc.?

2. How optimization was done with FFT/iFFT involved in MM-GPVAE?

---

> ### Author Response · Authors · 2023-11-18
>
> We thank the reviewer for the positive appraisal of the manuscript. We address some of the concerns and misunderstandings below.
> > The method proposed in this paper is combination of existing methods, not fundamentally different.
>
> We agree with the reviewer that the core contribution of this work represents a combination of GPFA, and GP-VAEs along linear partitioning of a latent space for datasets into shared and independent components of two modalities, and so is in a sense a combination of existing methods. However, we believe that we are the first time-series latent variable model developed that uses this partitioning to separate both shared and independent latent trajectories in multi-modal settings in neuroscience, and we do so in a way that is more flexible and scalable compared to competing multi-modal methods (see appendix). We feel that this is therefore a novel approach with wide applicability across a range of multi-modal neuroscientific experimental set-ups that we demonstrate in the manuscript. We also would like to highlight our novel amortization method in the Fourier domain for a deep GP model, as well as our derivation of a model specific ELBO in the appendix that precludes the need to approximate two of the terms in the objective.
> > More baselines, similar methods such as [1,4]
>
> [1] and [4] are not multi-modal models, and so we do not believe that they are fundamentally similar to our approach. We do, however, recognize that [1] and [4] each relate to one piece of our model, and we agree with the reviewer that comparisons of our multi-modal model to these unimodal variants will help us understand model performance. However, our comparisons of GPFA and GP-VAE in figures 1 and 3 are identical to the relevant variant of the method outlined in [1] and closely related to that of [4]. ELBO from [4] takes a similar general form to the GP-VAE ELBO we adapted which was presented in Casale et al 2018, barring a different observation likelihood and differently factorized variational distribution. Therefore, we consider our comparison to Casale et al 2018 to be a very close approximation to the comparison in [4]. We thank the reviewer for pointing us to this work and will add the citation of [4] and discussion as to this point in our revised version.
> > More baselines
>
> We benchmark against two prominent multi-modal models used for neuroscience in the appendix - one of these uses ODEs to characterize the latents, and the other uses an RNN. If the reviewer finds these benchmarks insufficient, please let us know.
> > Ablation study with fourier frequencies
>
> Thank you for this suggestion, we now address this in the global response.
>
>  > Interpretability
>
> We are very specific about interpretability in this work. In the synthetic examples, the "interpretability” of our model refers to the fact that we can recover latents that identify two features of the data that are meaningful to us, in this case, the digit scaling factor and angle. In the case of the real data examples, interpretability means the identified latent in the data corresponds to something scientifically meaningful - for example, the position of a visual stimulus, the beating pattern of a hawkmoth wing (Fig 5) , the behavioral state of a fly (Fig 4) , or the reaching position of a monkey (Fig 2 in appendix). We will edit our wording to be more specific about our use of the term ‘interpretability'
>
> ### Questions
> > 1
>
> For the real neural dataset examples in the main paper, the dimensionality was chosen for the neural dataset using PCA (selected for capturing greater than 97% of the variance) and the dimensionality was chosen for the behavioral data by increasing the number of latent dimensions until cross-validated performance did not show significant improvement. To determine the shared dimensionality, we similarly systematically increased the number of dimensions shared between datasets until predictive performance no longer improved. However, the results are not particularly sensitive to latent dimensionality within a few dimensions of either modality.
>
> > 2
>
> The encoder networks first identified latent variables per time-point, and was then passed through a FFT and reduced in dimensionality across time before being used to sample for the ELBO. Thus, inference was done in the Fourier domain. When decoded, the latents were mapped back out of the Fourier domain using an inverse transform. Because each of these transformations are linear operations, they are simply parts of the encoder and decoder networks. Network schematics for the exact mappings for each of the analyses can be found in figures 9-12 of the appendix. Please let us know if the reviewer needs any further clarification.
>
> We again thank the reviewer for their thoughtful comments and suggestions, and for their time reviewing the paper. If there are still any questions or concerns, we would be happy to clarify them and engage in further discussion.

---

> ### Author Response · Authors · 2023-11-19
> **Cont. for [1] and [4]**
>
> > More baselines, especially similar methods such as [1,4] need to be compared
>
> To elaborate further, the model in [1]  is a unimodal model developed for neural data that explores trial-by-trial deviations in trail-based neural data with exact repeated stimulus conditions, so alone the model introduced in [1]  is not directly applicable to our setting. However, we recognize that a “signal only” version of [1]  is an equivalent model to the poisson GPFA model first introduced in Zhao and Park 2017 learned with a Fourier representation, similar to our approach in an exclusively linear setting without deep amortization. Therefore, this version of [1] is an identical model to the “GPFA only” column in figure 3 in our paper. So, while we did not make this connection explicit in our paper, we do have a direct comparison here to the relevant modified version of [1]. (the only difference is our model uses amortized inference, instead of a black-box VI approach used in [1], which improves out-of-sample performance and allows us to adapt this representation to the deep setting). We will make sure this comparison to this version of a unimodal poisson-GPFA is clearer upon revision.
>
> Similarly, [4] is a unimodal GP-VAE model, and it uses Bernoulli likelihood and different factorized variational distribution but otherwise is closely related to the ELBO of Casale et al. Our primary contribution is comparing this same ELBO with and without the Fourier domain representation of the prior and latents (The prior and latents of [4] are identical to that of Casale et al.)

---

> ### Comment · Reviewer_Kdv4 · 2023-11-22
>
> I would like to thank the authors for the effort to address some of the questions. I am wondering for a given new dataset, how much tuning time would be needed if latent dimension, frequency pruning, and other hyperparameters were involved with cross validation, etc.
>
> Regarding the baseline comparison, I still felt that the model itself has been proposed or very similar to the existing ones. If the authors would like to emphasize their latent space partitioning idea, it may be important to compare with multi-modal feature fusion and other similar multi-modal embedding methods.
>
> I am little bit confused with the authors' statement: "highlight our novel amortization method in the Fourier domain for a deep GP model, as well as our derivation of a model specific ELBO in the appendix that precludes the need to approximate two of the terms in the objective." If the model has to be trained in the Fourier domain, is this what have to be done? It would be great if the authors can provide more details why the amortization methods in the Fourier domain is 'novel'.

---

> > ### Author Response · Authors · 2023-11-22
> >
> > >  [...] for a given new dataset [...]
> >
> > In addition to our simulated example, we tested MM-GPVAE with three neuroscience datasets ranging from limb tracking to stimulus movement and were able to adjust the model for each of these datasets very easily. In short, as long as we know the modalities we are working with and the size of them, we can initialize our $\sigma^2$ parameter as described in the appendix. Because inference generally only takes a few minutes to a few tens of minutes on a modern laptop, cross validation is not particularly time consuming and only needs to be performed once.
> >
> > > Regarding the baseline comparison [...]
> >
> > We compare to an RNN and ODE based multi-modal time series models in neuroscience in the appendix and show the differences in performance between these existing methods. Though there are other multi-modal models in the literature, we aren’t aware of other ones that include dynamics, and are specifically focused on neuroscience latent analysis. If the reviewer is aware of such models,  please let us know and we will happily include a comparison.
> >
> >
> > > [...] It would be great if the authors can provide more details why the amortization methods in the Fourier domain is 'novel' [...]
> >
> > Though a Fourier representation of the GP is used and is standard in the GP literature, as far as we know they have never been implemented in a deep setting, which requires not just a new parameterization of the nonlinear model but a distinct inference approach. This new inference approach (the amortization of Fourier frequencies) in our model differs from other GP models as well as other VAE models. To be clear, the contribution is that we encode a time-domain representation, and then transform it into the Fourier space prior to decoding back into the time domain. This extra encoding and decoding stages markedly scales and stabilizes inference despite being an additional linear step as it provides the GPVAE with a strong regularization.
> > This also allows us to modify the existing GPVAE to accurately identify the latent space, as evidenced by our simulated examples. This type of pruned variational posterior and amortized encoding implementation would not be readily available elsewhere or in any standard GP packages.
> >
> > The model inference does not have to be done in the Fourier domain - we have a boolean indicator for Fourier or time domain inference in our code. Though we highly encourage practitioners to use the Fourier representation as latent identification is markedly improved.
> >
> > We hope that we were able to sufficiently address all of the concerns and questions raised by the reviewer. If this is the case, we hope the reviewer could consider updating their score accordingly. If there are still any questions or concerns, we would be happy to clarify them and engage in further discussion.

---

### Author Response · Authors · 2023-11-18
**Global Response**

We thank the reviewers for their constructive comments. It seems most reviewers had a positive appraisal of our work, especially when it came to writing, clarity, and impact on the community. However, there were some consistent requests for clarification and additional evaluations.  They were 1) further evaluation on the use of frequency pruning in the Fourier domain, 2) clarification on how we selected dimensionality on our real-world evaluations, and 3) additional benchmarking. We address each in our global response and have included several new evaluations in the appendix.

### 1) Evaluating pruning in the Fourier domain.

We found when exploring early versions of our model that we needed to prune the frequency to accurately identify the latents. The Fourier representation alone, while it precluded the need of inverting the prior covariance matrix and offered some scaling improvements (like those seen in Hensmen et al 2017 and Keeley et al. 2020), was significantly more important when adapted to the GPVAE setting to identify the underlying latents. In fact, not pruning the frequencies yielded identified latents that looked nearly identical to the time-domain representation. We have added a figure in the appendix (Fig 6) showing how if we do not prune the frequencies in the MM-GPVAE, inference fails to be able to recover the true underlying latents (inference is also highly unstable in this setting). We are working to add an ‘unpruned’ condition to figure 1 in the paper, which explores the unimodal GPVAE, and are hoping to complete this as soon as possible. This will surely be ready upon a full revision. The latents here, again, when not pruned, are stochastic and look much like the standard VAE inferred latents with similar reconstruction error.

### 2) latent dimensions

A few reviewers asked for clarification as to how we identify the latent dimensionalities for our real-world evaluations. This was briefly discussed in the appendix but the treatment was cursory so we understand the reviewers confusion on this point. We explain our process in response to the individual reviewers below and have added a section in the appendix specifically outlining our cross-validation procedure for latent dimensionality identification on page 9 in the appendix. In short, we used an exhaustive cross-validation procedure for latent dimensionality identification. For more information, see reviewers responses and the current appendix version.

### 3) other benchmarking to existing models

Some reviewers asked for additional benchmarking of the MMGPVAE. While we compare the MMGPVAE to its unimodal variants (GPFA and GPVAE) in figure 3 in the main manuscript, we understand that reviewers may not consider this to be a complete set of benchmarks for this work. We agree with this assessment. We included additional evaluations compared to other multi-modal time-series models in the appendix, but it seems like we did not highlight these evaluations clearly enough in the manuscript. In short, we compared to an RNN based multi-modal model (Hurwitz et al 2021), as well as an ODE based multi-modal model (Sani et al 2021) in the appendix.  We find that these models fail when applied to our simulated data, as they do not have sufficient flexibility to describe data of both modalities. We perform a second comparison of the MMGPVAE to these existing models in a setting where all models are able to reconstruct the data of both modalities well, as a way to fairly compare the latent characterizations across all models. Please see the appendix, page 1 and 2,  for this discussion and comparison to these additional multi-modal approaches.

We made a choice in our main manuscript to focus on real-world applications of our model, focusing on a fly behavioral experiment and a hawkmoth visual-stimulus tracking experiment. This unfortunately came at the expense of highlighting this additional benchmarking in the main body of the paper.

### 4) additional evaluations

Reviewer bxWt made a number of good suggestions for some additional evaluations of the MMGPVAE. We have included these in the appendix. Please see reviewer bxWt’s comments and our response for more information.


Note that we are still in the process of incorporating other reviewer suggestions into the main manuscript, and we will be sure to include all the changes in the final manuscript. For now, we have addressed the most significant reviewer concerns in our current appendix, located in the supplementary zip file, and addressed reviewer concerns to the best of our ability in our responses below.


We hope that the revisions and responses below have sufficiently addressed all of the concerns and questions raised by the reviewers. If this is the case, we hope the reviewers could consider updating their score accordingly. If there are still any questions or concerns, we would be happy to clarify them and engage in further discussion.

---

### Meta-Review · Area_Chair_pKk4 · 2023-12-13

**Metareview:**

The authors propose a GP-VAE specialized for discovering latent structure in multi-model datasets in neuroscience involving simultaneous measurements of neural activity and behavior. The paper introduces a novel partitioning of the latent space, which is novel in the context of GPFA/GPVAE. The reviewers expressed some concerns that the novelty is confined to the partitioning scheme, but other reviewers pointed to the important need for such multi-modal approaches in neuroscience.

**Justification For Why Not Higher Score:**

Concerns of overall novelty beyond multi-modal data modeling and neuroscience, concerns of overly simplistic synthetic dataset evaluations.

**Justification For Why Not Lower Score:**

Novelty of GPVAE based modeling of multi-modal datasets, especially for neuroscience.

---

### Decision · Program_Chairs · 2024-01-16

Accept (poster)